# Score-Based Denoising Diffusion Models for Photon-Starved Image Restoration Problems

**Savvas Melidonis**                                                                 *s.melidonis@fz-juelich.de*
*Juelich Supercomputing Centre*
*Forschungszentrum Juelich*

**Yiming Xi**                                                                        *y.xi-7@sms.ed.ac.uk*
*University of Edinburgh*
*Heriot-Watt University*
*Maxwell Institute for Mathematical Sciences*

**Konstantinos C. Zygalakis**                                                        *k.zygalakis@ed.ac.uk*
*University of Edinburgh*
*School of Mathematics*
*Maxwell Institute for Mathematical Sciences*

**Yoann Altmann**                                                                    *y.altmann@hw.ac.uk*
*Heriot-Watt University*
*School of Engineering and Physical Sciences*
*Institute of Signals, Sensors and Systems*

**Marcelo Pereyra**                                                                  *m.pereyra@hw.ac.uk*
*Heriot-Watt University*
*School of Mathematical and Computer Sciences*
*Maxwell Institute for Mathematical Sciences*

**Reviewed on OpenReview:** *https://openreview.net/forum?id=UYXPt7HUdl*

## Abstract

Score-based denoising diffusion models have recently emerged as a powerful strategy to solve image restoration problems. Early diffusion models required problem-specific training. However, modern approaches can combine a likelihood function that is specified during test-time with a foundational pretrained diffusion model, which is used as an implicit prior in a Plug-and-Play (PnP) manner. This approach has been shown to deliver state-of-the-art performance in a wide range of image restoration problems involving Gaussian and mild Poisson noise. With extreme computer vision applications in mind, this paper presents the first PnP denoising diffusion method for photon-starved imaging problems. These problems arise in new quantum-enhanced imaging systems that exploit the particle nature of light to exceed the limitations of classical imaging. The problems involve highly challenging noise statistics, such as binomial, geometric, and low-intensity Poisson noise, which are difficult because of high uncertainty about the solution and because the models exhibit poor regularity properties (e.g., exploding scores, constraints). The proposed method is demonstrated on a series of challenging photon-starved imaging experiments with as little as 1 photon per pixel, where it delivers remarkably accurate solutions and outperforms alternative strategies from the state-of-the-art.

# 1   Introduction

Modern computer vision systems are increasingly required to operate in extreme conditions (e.g., ultra-fast acquisition times, low illumination, long-range, unconventional environments). This has led to the development of new quantum-enhanced imaging systems that exploit the particle nature of light to exceed the limitations of classical imaging strategies (Gibson et al., 2020).

The measurements produced by quantum-enhanced cameras are photon-starved and exhibit challenging noise statistics, such as binomial, geometric, and low-intensity Poisson noise (Altmann et al., 2017a;b). The top row of Figure 1 depicts an example of an urban scene as acquired by three different imaging systems involving Poisson, binomial, and geometric noise. Poisson-distributed measurements arise from the use of single-photon detectors capable of counting individual photons within a given time period (Altmann et al., 2017b; Rapp & Goyal, 2017; Shin et al., 2015; Ma et al., 2017) or standard charge-coupled device and complementary metal oxide semiconductor cameras (Duarte et al., 2008) that operate under poorly illuminated conditions. Measurements with binomial and geometric statistics arise from single-photon detectors (that is, detectors that are not able to accurately quantify photons beyond the first detection) such as quantum imaging sensors (Fossum et al., 2016; Ma et al., 2019) and single-photon avalanche diodes (Altmann et al., 2017b; Eisaman et al., 2011; Kirmani et al., 2014; Rapp & Goyal, 2017).

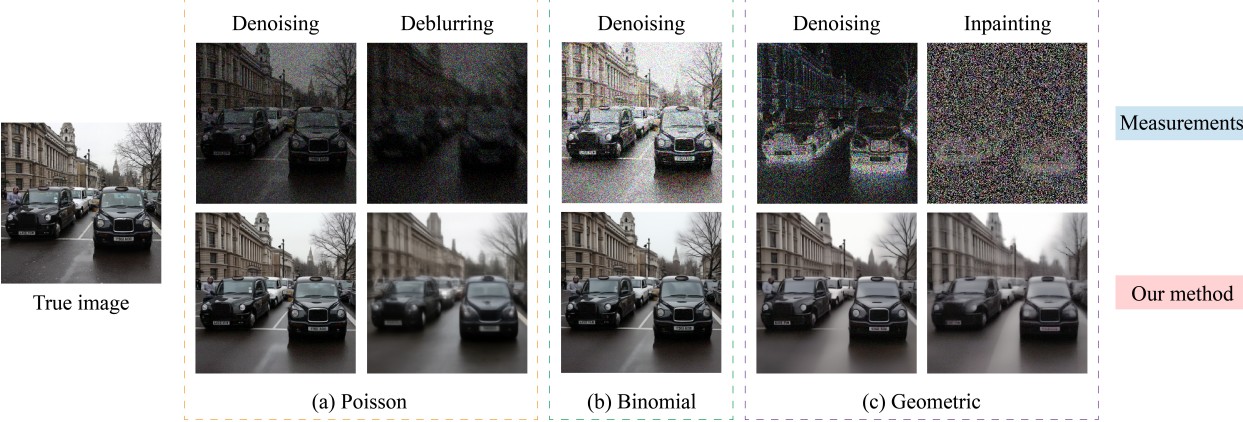

Figure 1: Restoration examples of our method: ground truth, measurements, and restored images for five image restoration tasks involving Poisson, binomial, and geometric noise.

Performing image restoration in problems involving photon-starved measurements is difficult and requires the development of specialised techniques. Early methods for low-photon imaging performed image restoration by minimising an energy function composed of a sum of a data term and a handcrafted prior (Oh et al., 2014; Chouzenoux et al., 2015; Figueiredo & Bioucas-Dias, 2010; Harmany et al., 2012; Setzer et al., 2010; Rapp & Goyal, 2017; Peng et al., 2020). Such methods are able to capture the structural properties of the image but struggle to restore fine detail. More recent methods are largely based on deep learning techniques. End-to-end strategies train a neural network to take as input the photon-starved measurements and return a reconstructed image (Sanghvi et al., 2022; Al-Shabili et al., 2022; Choi et al., 2018; Chi et al., 2020; Huang et al., 2023; Kang et al., 2023). Although such strategies can achieve remarkable performance in controlled environments, they are problem-specific and difficult to deploy flexibly. Alternatively, the so-called Plug-and-Play (PnP) approaches combine a pre-trained data-driven prior encoded by a deep neural network, with an explicit data fidelity term that is specified during test time (Hurault et al., 2024; Ryu et al., 2019; Marais & Willett, 2017; Rond et al., 2016; Cai et al., 2023; Luo et al., 2024). In particular, PnP approaches based on denoising diffusion models deliver unprecedented performance in problems involving Gaussian noise or mild Poisson noise through Gaussian approximation (Chung et al., 2023). However, for problems involving severe Poisson noise or binomial and geometric noise as discussed in this paper, the Gaussian approximation proves highly inaccurate and leads to poor image reconstructions with PnP methods based on denoising diffusion models (see Section 4).

This paper presents the first PnP denoising diffusion method suitable for photon-starved image restoration problems arising in quantum-enhanced computer vision. The proposed method combines ideas of Bayesian inference and optimisation to provide accurate restoration results in a highly computationally efficient manner. In particular, contrary to existing state of the art methods, our approach doesn't employ the Gaussian approximation for the likelihood but instead makes appropriate use of the true one. The proposed method is demonstrated on a series of challenging photon-starved imaging problems and comparisons with alternative approaches from the state-of-the-art (see Fig. 1). These experiments are carried out on the FFHQ (Karras et al., 2019) and ImageNet (Deng et al., 2009) datasets.

## 2 Background

### 2.1 Low-Photon Imaging

We consider the recovery of an unknown image $x^\star \in \mathbb{R}^d$ from a set of photon-starved measurements $y \in \mathbb{R}^m$, related to $x^\star$ by a statistical model with likelihood $x \mapsto p_0(y|x)$. Canonical examples include image restoration problems involving low-photon Poisson, binomial, or geometric noise. In the case of Poisson noise, $y = [y_1, \ldots, y_m] \in \mathbb{N}_0^m$ consists of discrete photon counts, modelled as a realisation of the Poisson distribution

$$\mathbf{y}|x^\star \sim \mathcal{P}(\alpha \cdot \mathcal{H}(x^\star)), \tag{1}$$

where $\mathcal{H}$ is a non-negative linear operator modelling deterministic aspects of the imaging system (often distortions such as non-uniform gain, blurring operator etc.) and $\alpha > 0$ is related to the strength of shot noise (the larger $\alpha$, the easier the image restoration problem). The respective negative log-likelihood $f_y(x) \triangleq -\log p_0(y|x)$ is given by

$$f_y(x) = \sum_{i=1}^m \left[(\alpha \cdot \mathcal{H}(x))_i - y_i \log((\alpha \cdot \mathcal{H}(x))_i) + \log(y_i!)\right] , \tag{2}$$

where all $y_i$ are conditionally independent given $x$. While the Poisson noise model is widely adopted to model random photon counts, single-photon detectors can result in more complex observation models when are utilised at the edge of their ideal working conditions (Altmann et al., 2017b; Eisaman et al., 2011; Kirmani et al., 2014; Rapp & Goyal, 2017; Fossum et al., 2016; Ma et al., 2019). This is typically the case for detectors such as single-photon avalanche diodes that need to be reset after each photon detection, leading to dead-time during which incoming photons can be missed. If the detectors are reset at regular time intervals (or acquisition period), the measurement image $y = [y_1 \ldots, y_m] \in \mathbb{N}_0^m$ is better modelled as a set of sums of binary detections $\{0, 1\}$ or Bernoulli random variables, that is,

$$\mathbf{y}|x^\star \sim \mathcal{B}in\left(t, 1 - e^{-\alpha \cdot \mathcal{H}(x^\star)}\right) , \tag{3}$$

where $\mathcal{B}in(\cdot, \cdot)$ stands for the product of independent binomial distributions, $t \in \mathbb{N}$ is the number of repetition periods for each pixel, $\mathcal{H}$ is again a non-negative linear operator, and $\alpha > 0$ is an efficiency parameter that controls the probability of successful detection. Note that Eq. 1 is a good approximation of Eq. 3 in practice if $\alpha$ in Eq. 3 is sufficiently small. The respective negative log-likelihood is given by

$$f_y(x) = \sum_{i=1}^m \left[-y_i \log(1 - e^{-(\alpha \cdot \mathcal{H}(x)_i)}) + \alpha \cdot (t_i - y_i)(\mathcal{H}(x))_i\right] . \tag{4}$$

Lastly, in first-photon imaging systems (Kirmani et al., 2014; Liu et al., 2017; Peng et al., 2020), the detector records the number of acquisition periods, or repetitions, until the first photon detection, for each pixel individually. Then, $y = [y_1 \ldots, y_m] \in \mathbb{N}_0^m$ is related to $x^\star$ by

$$\mathbf{y}|x^\star \sim \mathcal{G}eo\left(1 - e^{-\alpha \cdot \mathcal{H}(x^\star)}\right) , \tag{5}$$

where $\mathcal{G}eo(\cdot)$ stands for the product of independent geometric distributions. The associated negative log-likelihood is

$$f_y(x) = \sum_{i=1}^m \left[(y_i - 1)(\alpha \cdot \mathcal{H}(x))_i - \log\left(1 - e^{-(\alpha \cdot \mathcal{H}(x))_i}\right)\right] . \tag{6}$$

## 2.2 Score-Based Denoising Diffusion Models

Score-based denoising diffusion models are a state-of-the-art generative modelling strategy to draw samples from a target distribution $\pi$ on $\mathbb{R}^d$, known only through a sample $\{x_i\}_{i=1}^M$. The models are constructed from a stochastic transport map between a $d$-dimensional Gaussian distribution $\mathcal{N}(0, \mathbb{I}_d)$ and $\pi$. In particular, for an appropriate choice of the drift $h$ and diffusion coefficient $g$, solutions of the stochastic differential equation (SDE) as satisfied below

$$d\mathbf{x}_t = h(\mathbf{x}_t, t)dt + g(t)d\mathbf{w}, \quad \mathbf{x}_0 \sim \pi, \tag{7}$$

satisfy that $\mathbf{x}_t \sim \mathcal{N}(0, \mathbb{I}_d)$ as $t \to \infty$. Here $\mathbf{w}$ is the $d$-dimensional Brownian motion. The reverse-time SDE associated to Eq. 7 is given by (Anderson, 1982)

$$d\mathbf{x}_t = [h(\mathbf{x}_t, t) - g(t)^2 \nabla \log p_t(\mathbf{x}_t)]dt + g(t)d\bar{\mathbf{w}}, \quad \mathbf{x}_\infty \sim \mathcal{N}(0, \mathbb{I}_d) \tag{8}$$

where $p_t(\mathbf{x_t})$ is the marginal density of $\mathbf{x}_t$ at time t, $\bar{\mathbf{w}}$ is a $d$-dimensional Brownian motion, and now $t$ flows backwards from infinity to $t = 0$. Score-based denoising diffusion models use Eq. 7 together with weighted score-matching techniques (Hyvärinen & Dayan, 2005; Vincent, 2011) and specialised architectures to learn an accurate neural-network approximation of the score function $s \mapsto \nabla \log p_t(s)$ (Song et al., 2021). New samples from $\pi$ are then generated by using the estimated scores to approximately solve Eq. 8. This is achieved by using a discrete-time approximation of Eq. 8 initialised with $\mathbf{x}_T \sim \mathcal{N}(0, \mathbb{I}_d)$ for a large finite $T$. A popular specific choice of Eq. 7-8 is

$$d\mathbf{x}_t = -\frac{1}{2}\beta(t)\mathbf{x}_t dt + \sqrt{\beta(t)}d\mathbf{w}, \tag{9}$$

$$d\mathbf{x}_t = \left[-\frac{1}{2}\beta(t)\mathbf{x}_t - \beta(t)\nabla \log p_t(\mathbf{x}_t)\right]dt + \sqrt{\beta(t)}d\bar{\mathbf{w}}_t, \tag{10}$$

where $t \mapsto \beta(t) \in \mathbb{R}_0^+$ is a scheduling function. An Euler-Maruyama approximation of Eq. 8 for the specific choice Eq. 10 leads to the popular Denoising Diffusion Probabilistic Model (DDPM) (Ho et al., 2020). A more sophisticated approximation of this backward process leads to the Denoising Diffusion Implicit Model (DDIM) (Song et al., 2020), which requires significantly less discretisation steps than DDPM and is therefore more computationally efficient (as it requires less neural function evaluations).

## 2.3 Image Restoration with Denoising Diffusion Models

Following the remarkable success of denoising diffusion models in generative modelling tasks, there has been a great interest in adapting them for image restoration. In principle, this can be achieved by modifying Eq. 8 to use the conditional score $s \mapsto \nabla \log p_t(s|y)$. However, the resulting diffusion models are problem-specific and cannot be deployed flexibly. This has led to significant research on strategies that combine a foundational pre-trained DDPM or DDIM prior, with an explicit likelihood function specified during test time. In particular, the conditional score function can be decomposed as $\nabla \log p_t(x_t|y) = \nabla \log p_t(x_t) + \nabla \log p_t(y|x_t)$, where the score $\nabla \log p_t(x_t)$ is encoded in a DDPM or DDIM targeting $\pi$. However, evaluating the likelihood score $\nabla \log p_t(y|x_t)$ requires computing the intractable integral $p_t(y|x_t) = \int p_0(y|x_0)p_t(x_0|x_t)\mathrm{d}x_0$, which is not feasible even if the likelihood $p_0(y|x_0)$ is known.

To address this difficulty, a number of different approaches have been proposed. Notably, diffusion posterior sampling (DPS) (Chung et al., 2023) introduces a guidance term to modify a foundational DDPM so as to target $(\mathbf{x}_0|\mathbf{y} = y)$ instead of $\mathbf{x}_0$. More precisely, each iteration of DDPM is corrected by a gradient step along $\nabla_{\mathbf{x}_t} \log p_0(y|\hat{\mathbf{x}}_0(\mathbf{x}_t))$ that promotes consistency with $y$, where $\hat{\mathbf{x}}_0(\mathbf{x}_t)$ is an estimate of the mean of $(\mathbf{x}_0|\mathbf{x}_t)$ derived from the scores underpinning the DDPM through Tweedie's formula. Though DPS can deliver state-of-the-art results, it requires extensive fine-tuning and has a high computational cost. The pseudoinverse-guided diffusion model ($\Pi$GDM) (Song et al., 2022) and the denoising diffusion restoration model (DDRM) (Kawar et al., 2022) improve on DPS by adopting a DDIM strategy that is significantly more computationally efficient, and improving the approximation of $\nabla \log p_t(y|x_t)$ for the specific case of Gaussian likelihood functions. An alternative recently proposed approach (Cardoso et al., 2024) exploits a sequential Monte

Carlo strategy to deal with the intractable score $\nabla \log p_t(y|x_t)$. However, the aforementioned approaches (Kawar et al., 2022; Song et al., 2022; Cardoso et al., 2024) cannot be easily extended to the low-photon image restoration problems considered in this paper.

Denoising diffusion models for plug-and-play image restoration (DiffPIR) (Zhu et al., 2023) modifies a foundational DDIM by introducing a half-quadratic relaxation to decouple the likelihood term and the prior. This relaxation leads to a modified DDIM whose iterations are guided by a least-squares correction step that promotes consistency with $y$, with the weight of the guidance increasing across iterations. DiffPIR is significantly more robust than DPS and delivers state-of-the-art performance in less iterations than competing DDIM strategies. Similarly, plug-and-play split Gibbs sampler (PnP-SGS) (Coeurdoux et al., 2023) also uses half-quadratic splitting to modify a DDIM foundational prior, but relies on a stochastic sampling step to introduce consistency with $y$ rather than a gradient step, leading to a DDIM-within-Gibbs sampler.

## 3 Proposed Method

Our proposed method is a generalisation of DiffPIR to non-Gaussian likelihood functions that are log-concave and potentially non-smooth. This class of likelihoods includes the Poisson, geometric, and binomial models considered above, which are log-concave and have non-negativity constraints as well as non-Lipschitz gradients. For presentation clarity, we recall DiffPIR in Eq. 1 (for Gaussian likelihoods), where $\beta_t$ corresponds to a time discretization of the scheduling function $\beta(t)$, $\alpha_t = 1 - \beta_t$, $\bar{\alpha}_t = \prod_{s=1}^{t} \alpha_s$, $s_\theta$ is the score function of the implicit prior, $\lambda$ is a regularization controlling the balance between the foundational DDIM and the guidance term, and $\zeta \in (0, 1)$ controls the stochasticity of the DDIM kernel, and $\sigma$ is the level of the Gaussian noise in $y$.

---

**Algorithm 1** DiffPIR

**Require:** $\{\bar{\alpha}_t\}_{t=1}^T, s_\theta, T, y, \sigma, \zeta, \lambda$
1: $x_T \sim \mathcal{N}(0, \mathbb{I}_d)$
2: **for** $t = T : 1$ **do**
3: $\quad x_0^{(t)} = \frac{1}{\sqrt{\bar{\alpha}_t}} \left( x_t + (1 - \bar{\alpha}_t) \, s_\theta(x_t, t) \right), \quad \rho_t = \sigma \cdot \lambda \bar{\alpha}_t / (1 - \bar{\alpha}_t)$
4: $\quad \hat{x}_0^{(t)} = \arg\min_x \|y - \mathcal{H}(x)\|_2^2 + \frac{\rho_t}{2} \left\| x - x_0^{(t)} \right\|_2^2$
5: $\quad \hat{\epsilon} = \frac{1}{\sqrt{1 - \bar{\alpha}_t}} \left( x_t - \sqrt{\bar{\alpha}_t} \hat{x}_0^{(t)} \right)$
6: $\quad \epsilon_t \sim \mathcal{N}(0, \mathbb{I}_d)$
7: $\quad x_{t-1} = \sqrt{\bar{\alpha}_{t-1}} \hat{x}_0^{(t)} + \sqrt{1 - \bar{\alpha}_{t-1}} \left( \sqrt{1 - \zeta} \hat{\epsilon} + \sqrt{\zeta} \epsilon_t \right)$
8: **end for**
9: **Return** $x_0$

---

**Algorithm 2** Proposed generalisation of DiffPIR (ProxDiffPIR)

**Require:** $\{\bar{\alpha}_t\}_{t=1}^T, s_\theta, T, y, \zeta, \lambda$
1: $x_T \sim \mathcal{N}(0, \mathbb{I}_d)$
2: **for** $t = T : 1$ **do**
3: $\quad x_0^{(t)} = \frac{1}{\sqrt{\bar{\alpha}_t}} \left( x_t + (1 - \bar{\alpha}_t) \, s_\theta(x_t, t) \right), \quad \rho_t = \lambda \bar{\alpha}_t / (1 - \bar{\alpha}_t)$
4: $\quad \hat{x}_0^{(t)} = \text{prox}_{f_y^b}^{\rho_t} \left( x_0^{(t)} \right) = \arg\min_x f_y^b(x) + \frac{\rho_t}{2} \left\| x - x_0^{(t)} \right\|_2^2$
5: $\quad \hat{\epsilon} = \frac{1}{\sqrt{1 - \bar{\alpha}_t}} \left( x_t - \sqrt{\bar{\alpha}_t} \hat{x}_0^{(t)} \right)$
6: $\quad \epsilon_t \sim \mathcal{N}(0, \mathbb{I}_d)$
7: $\quad x_{t-1} = \sqrt{\bar{\alpha}_{t-1}} \hat{x}_0^{(t)} + \sqrt{1 - \bar{\alpha}_{t-1}} \left( \sqrt{1 - \zeta} \hat{\epsilon} + \sqrt{\zeta} \epsilon_t \right)$
8: **end for**
9: **Return** $x_0$

---

Note that DiffPIR is a standard DDIM procedure to draw samples from the prior, with the addition of a guidance step

$$\hat{x}_0^{(t)} = \arg\min_x \|y - \mathcal{H}(x)\|_2^2 + \frac{\rho_t}{2} \left\| x - x_0^{(t)} \right\|_2^2. \tag{11}$$

This step is equivalent to a proximal step for the Gaussian negative log-likelihood function (Bauschke et al., 2017), which suggests a natural generalisation of DiffPIR for the general class of likelihood functions discussed previously. That is, we could replace line 4 in Algorithm 1 with the proximal step associated with $f_y$, that is,

$$\hat{x}_0^{(t)} = \text{prox}_{f_y}^{\rho_t}\left(x_0^{(t)}\right) = \arg\min_x f_y(x) + \frac{\rho_t}{2}\left\|x - x_0^{(t)}\right\|_2^2. \tag{12}$$

where $f_y(x) = -\log p_0(y|x)$ is given, e.g., by Eq. 2, 4, 6 depending on the type of noise involved. Note that, unlike the Gaussian case, the solution of this optimization problem doesn't admit a closed-form solution and one needs to compute a solution iteratively by using an optimization algorithm. In addition, there are positivity constraints to be taken into account because the logarithmic terms $\log(\cdot)$ cannot be calculated when $\mathcal{H}(x) \leq 0$, and so unconstrained optimization tools cannot be directly applied. To address this difficulty, we introduce the approximation $f_y^b$ of $f_y$ given by $f_y^b = f(\mathcal{H}(x) + b)$, for some small positive $b$ obtaining the new optimisation problem

$$\hat{x}_0^{(t)} = \text{prox}_{f_y^b}^{\rho_t}\left(x_0^{(t)}\right) = \arg\min_x f_y^b(x) + \frac{\rho_t}{2}\left\|x - x_0^{(t)}\right\|_2^2. \tag{13}$$

When compared to Eq. 12, 13 allows the relaxation of the strict condition $\mathcal{H}(x) > 0$ to $\mathcal{H}(x) \geq 0$ (see Melidonis et al. (2023)). Problem 13 can then be solved with a generic solver that allows the method to be deployed flexibly. We use L-BFGS-B, which is a version of the L-BFGS method that can deal with non-negativity constraints. Of course, it is also possible to develop specialised schemes to solve Eq. 13 more efficiently for specific forms of $f_y^b$ and $\mathcal{H}$ of interest (Figueiredo & Bioucas-Dias, 2010).

The resulting generalisation of DiffPIR, namely ProxDiffPIR, is summarised in Eq. 2 below. We view Eq. 2 as an approximation of a DDIM targeting $(\mathbf{x}_0|\mathbf{y} = y)$ where, in the absence of the intractable score $\nabla \log p_t(y|x_t)$, we promote consistency with $y$ by using the unconditional score $\nabla \log p_t(x_t)$ together with a penalised least-squares correction step derived from the likelihood $p_0(y|x)$. When $f_y(x) = -\log p_0(y|x)$ is log-concave, this correction step is firmly nonexpansive (a weak form of contractivity). By comparison with DPS, which relies on an explicit gradient step that is not usually firmly non-expansive, the proposed method is more robust, as we illustrate in the experiments of Section 4.

## 4 Numerical Experiments

We illustrate the performance of the proposed method through experiments related to non-blind Poisson image deconvolution, binomial image denoising, and geometric image inpainting. For these experiments, we used 30 test images from the validation datasets of FFHQ $256 \times 256$ (Karras et al., 2019) and ImageNet $256 \times 256$ (Deng et al., 2009), and made use of pretrained diffusion models on the aforementioned datasets (Chung et al., 2023; Dhariwal & Nichol, 2021). For Poisson image restoration, we report comparisons with alternative score-based diffusion methodologies and other state-of-the-art techniques. Specifically, in all experiments, ProxDiffPIR (Algorithm 2) is compared with DPS (Chung et al., 2023) as implemented with the likelihood function $p_0(y|x)$ without a Gaussian approximation, henceforth denoted as DPS-t. In addition, for the Poisson and binomial experiments, we also report comparisons with DPS with a Gaussian likelihood approximation of the true likelihood function $p_0(y|x)$, as originally proposed by the authors in Chung et al. (2023). The implementation details of DPS and DPS-t are given in Appendix B. We also compare with the Plug-and-Play (PnP) ADMM scheme PIP (Rond et al., 2016), which uses a patch-based BM3D denoiser, the unrolled Plug-and-Play (PnP) network PhD-Net (Sanghvi et al., 2022), and the Bregman proximal gradient method, namely PnP-BPG (Hurault et al., 2024). For the binomial and geometric experiments, PIP is adapted by modifying the term associated with likelihood within the ADMM scheme as proposed in Melidonis et al. (2023). Lastly, for DPS and our method, rather than presenting a single sample as a solution, we generate $N$ samples and average them to reduce the variability of our estimates (in our experiments, we use $N = 3$ to achieve a compromise between accuracy and computing time). We assess reconstruction quality by computing peak signal-to-noise ratio (PSNR), structural similarity index measure (SSIM) (Wang et al., 2004) and learned perceptual image patch similarity (LPIPS) (Zhang et al., 2018). The finetuning details of ProxDiffPIR are provided in Appendix A. The experiments were run on an Intel(R) Core(TM)

i9-9940X CPU @ 3.30GHz with NVIDIA GeForce RTX 2080 Ti with 11GB memory by using `Pytorch`. For the LBFGS-B solver, we used the `scipy` library. The source code will be released upon paper's acceptance.

## 4.1 Non-blind Poisson image deconvolution

We aim to recover $x^\star$ from a noisy blurred observation $y \sim \mathcal{P}(\alpha \cdot \mathcal{H}(x^\star))$, where the blur operator $\mathcal{H}$ is known. In these experiments, we considered two forward operators $\mathcal{H}$; a truncated Gaussian blur kernel of size $9 \times 9$ with bandwidth 3-pixels and a motion blur kernel of size $9 \times 9$ (see supplementary material). We tested the performance of the methods at three different photon levels ($\alpha = 5, 10, 20$).

### 4.1.1 ImageNet

We now present the quantitative results for this experiment with the ImageNet dataset, which are summarized in Table 1. We observe that ProxDiffPIR is very competitive across all metrics considered, while DPS strategies can produce unstable results even after extensive finetuning (see Appendices B.1.3 and B.2.1, for more details). For example, DPS-t was numerically explosive for all images in the case of motion blur and for some images under Gaussian blur, while DPS was unstable for some experiments under photon levels $\alpha = 10$ and $\alpha = 20$. In the presence of high noise ($\alpha = 5$), DPS can be competitive in LPIPS but not in PSNR. In all other cases, we observe that ProxDiffPIR performs similarly or better than DSP in terms of LPIPS. We have noticed that the low performance of DPS in terms of PSNR is due to noisy reconstructions in many testing images, see Figure 2. This is also the case in the presence of medium noise ($\alpha = 10$); see Table 1 and Figure 2. In Figure 2, we also observe that alternative deterministic strategies cannot offer recoveries with fine details (which quantitatively leads to high LPIPS). In contrast, our method outperforms the DPS and deterministic methods by providing fine-detailed and smooth reconstructions. The illustration in Figure 2 is in agreement with the quantitative results reported in Table 1. In this experiment, our method requires around 37 seconds per image, while DPS-t and DPS need 520 seconds per image. See Table 5.

Table 1: Non-blind Poisson image deconvolution experiment: quantitative image restoration results (average over 30 images from the ImageNet validation dataset). For each quality metric, the best result is shown in bold and the second best is underlined.

| Noise Level | Kernel | Metrics | Ours | DPS-t | DPS | PnP-BM3D | PhD-Net | B-PnP |
|---|---|---|---|---|---|---|---|---|
| $\alpha = 5$ 
 high noise | Gaussian deblur | PSNR | **22.60** | 16.66 | 20.25 | 20.25 | 22.39 | 20.16 |
| | | SSIM | **0.60** | 0.26 | 0.56 | 0.55 | **0.60** | 0.58 |
| | | LPIPS | **0.42** | 0.60 | **0.42** | 0.54 | 0.52 | 0.51 |
| | motion deblur | PSNR | **22.86** | - | 21.01 | 22.05 | 22.44 | 20.23 |
| | | SSIM | **0.61** | - | 0.60 | 0.59 | 0.60 | 0.60 |
| | | LPIPS | 0.41 | - | **0.39** | 0.53 | 0.51 | 0.50 |
| $\alpha = 10$ 
 medium noise | Gaussian deblur | PSNR | **23.14** | 16.55 | 17.59 | 22.66 | 23.02 | 21.58 |
| | | SSIM | **0.63** | 0.24 | 0.30 | 0.61 | **0.63** | 0.62 |
| | | LPIPS | **0.39** | 0.62 | 0.56 | 0.51 | 0.49 | 0.48 |
| | motion deblur | PSNR | **23.75** | - | 21.56 | 22.82 | 23.13 | 21.72 |
| | | SSIM | **0.66** | - | 0.61 | 0.62 | 0.63 | 0.64 |
| | | LPIPS | **0.38** | - | **0.38** | 0.50 | 0.48 | 0.46 |
| $\alpha = 20$ 
 low noise | Gaussian deblur | PSNR | **23.63** | 17.87 | 18.15 | 23.20 | 23.52 | 21.41 |
| | | SSIM | **0.65** | 0.49 | 0.33 | 0.63 | **0.65** | 0.64 |
| | | LPIPS | **0.37** | 0.57 | 0.54 | 0.49 | 0.47 | 0.45 |
| | motion deblur | PSNR | **24.16** | - | 21.64 | 23.56 | 23.74 | 20.91 |
| | | SSIM | **0.66** | - | 0.60 | 0.65 | 0.65 | **0.66** |
| | | LPIPS | **0.36** | - | 0.40 | 0.47 | 0.45 | 0.44 |

### 4.1.2 FFHQ

The quantitative results for this experiment are summarized in Table 2. We observe that the diffusion-based strategies ProxDiffPIR and DPS are very competitive compared to deterministic alternatives in terms of all the metrics. This is in agreement with Figure 3, where we observe that ProxDiffPIR and DPS achieve very fine-detailed reconstructions compared to non-diffusion strategies. On the other hand, DPS-t leads to noisy reconstructions or to exploding gradients (see, for example, the results under a motion blur in Table 2). Among diffusion-based strategies, our method achieves the best performance in PSNR and SSIM, while DPS slightly outperforms our method in average LPIPS by a maximum difference of 0.02. We would note that LPIPS is a perceptual metric and thus has the tendency to favour images with fine details, even in

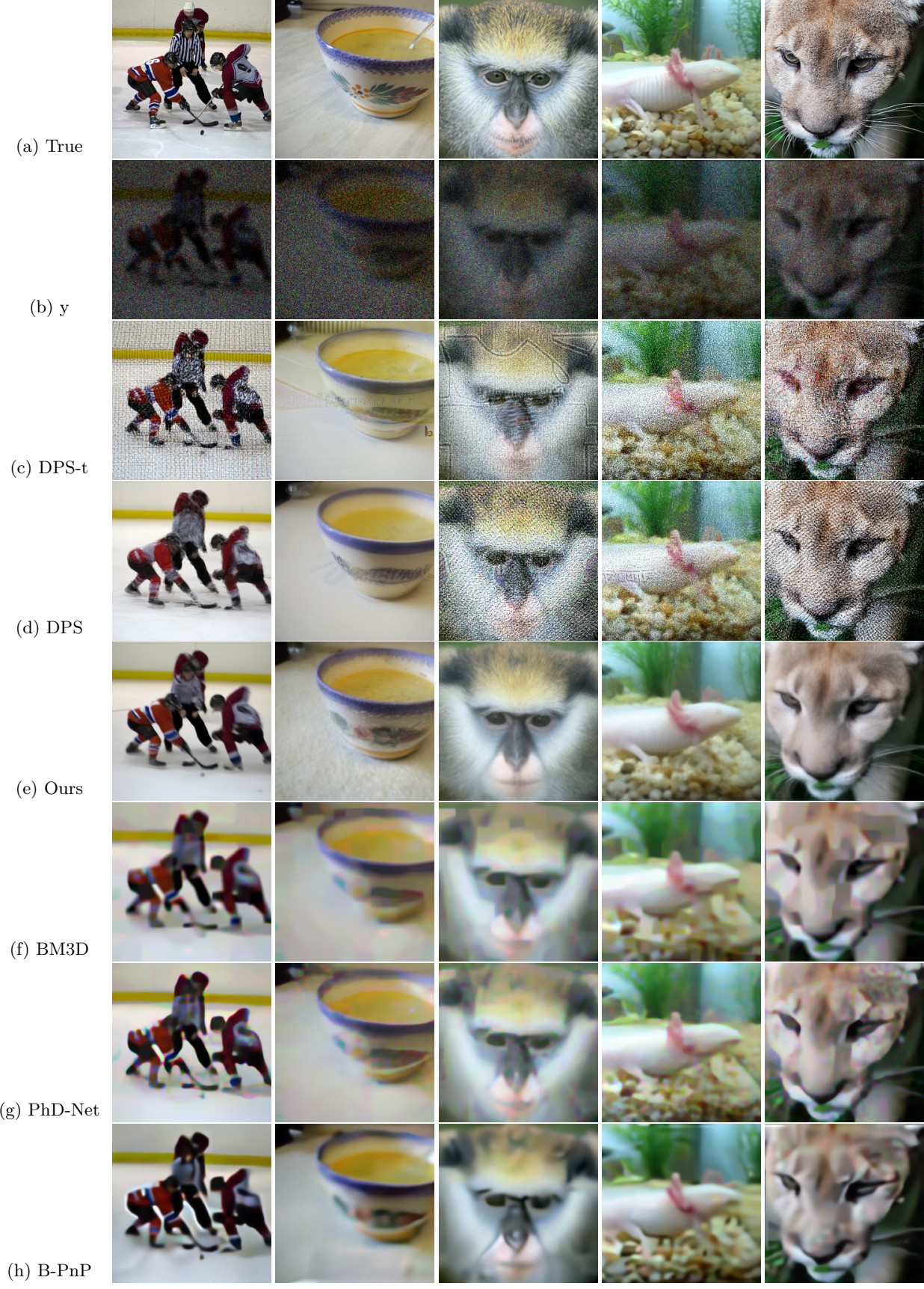

Figure 2: Poisson deblurring example ($\alpha = 5$ and $\alpha = 10$) with Gaussian blur on ImageNet dataset.

Table 2: Quantitative mean and standard deviation results over 30 images from the FFHQ validation dataset for the **Poisson deblurring** problem.

| Noise Level | Kernel | Metrics | ProxDiffPIR | DPS-t | DPS | BM3D | PhD-Net | B-PnP |
|---|---|---|---|---|---|---|---|---|
| $\alpha = 5$ high noise | Gaussian blur | PSNR | **24.77** | 20.35 | 22.55 | 23.27 | 23.72 | 21.21 |
| | | SSIM | **0.70** | 0.49 | 0.68 | 0.64 | 0.65 | 0.64 |
| | | LPIPS | 0.33 | 0.46 | **0.31** | 0.54 | 0.51 | 0.49 |
| | Motion blur | PSNR | **25.05** | - | 22.97 | 23.42 | 23.78 | 21.03 |
| | | SSIM | **0.72** | - | 0.70 | 0.65 | 0.65 | 0.65 |
| | | LPIPS | 0.31 | - | **0.30** | 0.53 | 0.49 | 0.48 |
| $\alpha = 10$ medium noise | Gaussian blur | PSNR | **25.12** | 19.94 | 23.65 | 24.20 | 24.50 | 22.79 |
| | | SSIM | **0.71** | 0.42 | 0.70 | 0.67 | 0.68 | 0.69 |
| | | LPIPS | 0.31 | 0.52 | **0.29** | 0.51 | 0.48 | 0.46 |
| | Motion blur | PSNR | **25.12** | - | 23.70 | 24.44 | 24.61 | 22.61 |
| | | SSIM | **0.72** | - | 0.71 | 0.68 | 0.69 | 0.69 |
| | | LPIPS | **0.29** | - | **0.29** | 0.49 | 0.45 | 0.45 |
| $\alpha = 20$ low noise | Gaussian blur | PSNR | **25.85** | 20.20 | 24.23 | 24.83 | 25.13 | 22.71 |
| | | SSIM | **0.74** | 0.40 | 0.71 | 0.70 | 0.71 | 0.71 |
| | | LPIPS | 0.29 | 0.52 | **0.28** | 0.45 | 0.45 | 0.43 |
| | Motion blur | PSNR | **26.50** | - | 24.78 | 25.13 | 25.32 | 22.00 |
| | | SSIM | **0.77** | - | 0.74 | 0.71 | 0.71 | 0.71 |
| | | LPIPS | 0.29 | - | **0.27** | 0.46 | 0.43 | 0.42 |

the presence of sharpening artifacts; we are therefore not overly concerned about minor variations in LPIPS performance. Moreover, it should be noted that DPS required highly computationally expensive fine-tuning to produce these results (see Appendices B.1.3 and B.2.1 for more details), and despite the fine-tuning DPS is not robust. For example, in Figures 3, we noticed that DPS fails to correctly estimate the brightness of the true image (leading to low PSNR values); this is due to the bias resulting from the use of an approximate likelihood function. Also note that DPS reconstructions exhibit some artifacts (see, for example, the last two columns in Figure 3). By comparison, our method consistently delivers strong results. In addition, for this experiment, our method required 23 seconds per image, whereas the DPS methods required 110 seconds. See Table 6.

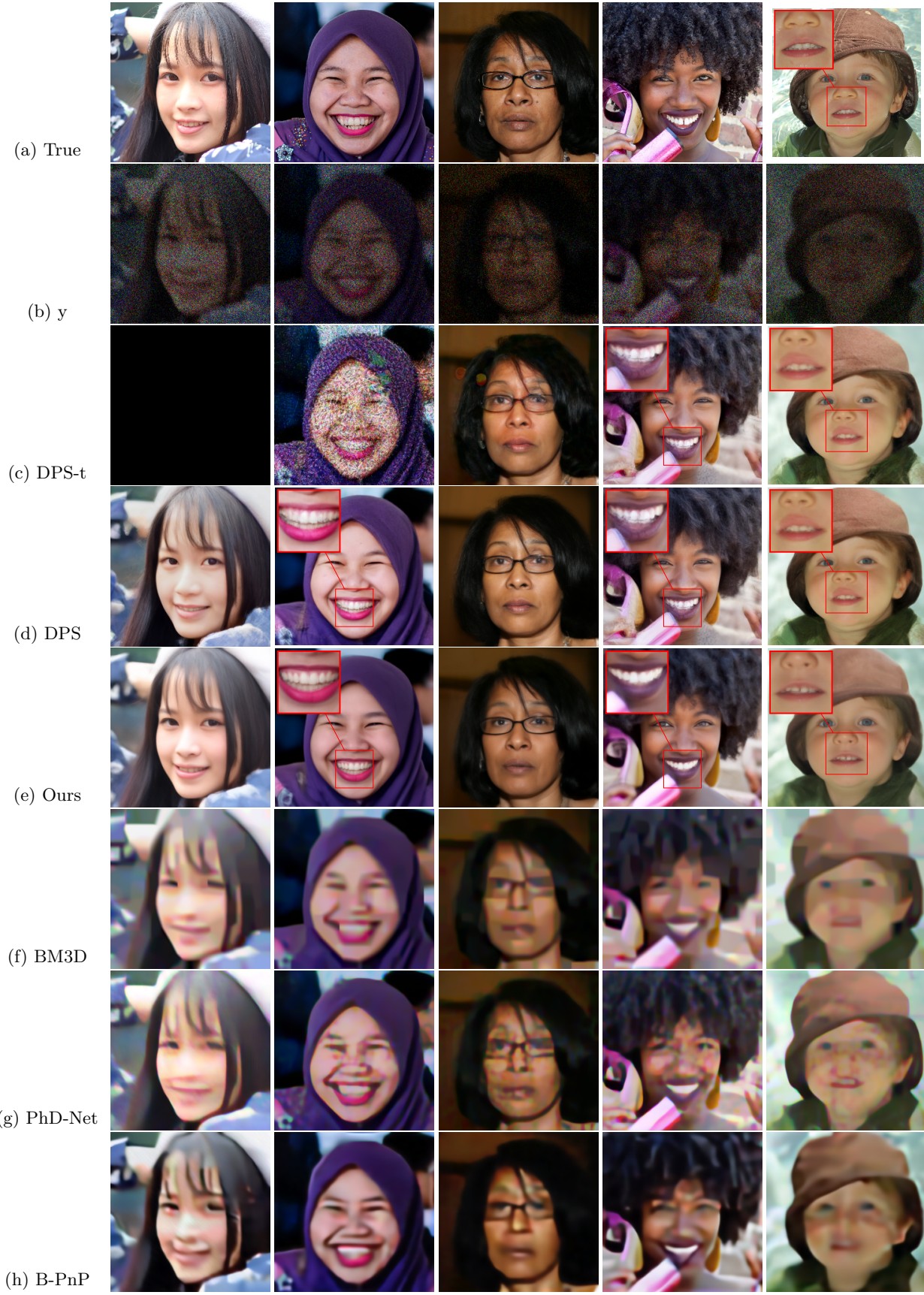

Figure 3: Poisson deblurring example ($\alpha = 5$) with motion blur on the FFHQ dataset.

## 4.2 Binomial denoising

We now aim to recover $x^\star$ from a noisy observation $y \sim \mathcal{B}in(t, 1 - e^{-\alpha \cdot x^\star})$. We consider two low-photon scenarios: $(\alpha = 2.5, t = 10)$ and $(\alpha = 0.25, t = 100)$. We present results with the ImageNet and FFHQ datasets, as summarized below.

Table 3: Binomial denoising experiment: quantitative image restoration results (averaged over 30 images from the FFHQ and ImageNet validation datasets). For each quality metric, the best result is shown in bold and the second best is underlined.

| Noise Level | Dataset | Metrics | Ours | DPS-t | DPS | PnP-BM3D |
|---|---|---|---|---|---|---|
| $\alpha = 2.5, t = 10$ | FFHQ | PSNR | **28.78** | - | - | 26.32 |
| | | SSIM | **0.85** | - | - | 0.76 |
| | | LPIPS | **0.22** | - | - | 0.41 |
| | ImageNet | PSNR | **27.97** | - | - | 25.29 |
| | | SSIM | **0.81** | - | - | 0.73 |
| | | LPIPS | **0.24** | - | - | 0.40 |
| $\alpha = 0.25, t = 100$ | FFHQ | PSNR | **29.83** | - | - | 26.96 |
| | | SSIM | **0.87** | - | - | 0.78 |
| | | LPIPS | **0.20** | - | - | 0.39 |
| | ImageNet | PSNR | **29.45** | - | - | 25.85 |
| | | SSIM | **0.85** | - | - | 0.76 |
| | | LPIPS | **0.20** | - | - | 0.37 |

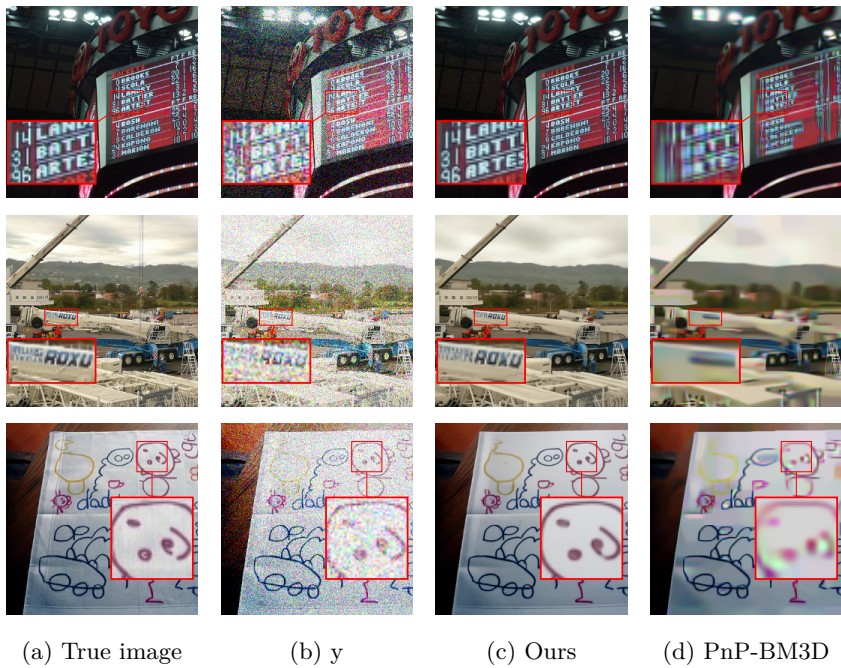

(a) True image  (b) y  (c) Ours  (d) PnP-BM3D

Figure 4: Binomial denoising example $(\alpha = 2.5, t = 10)$ on ImageNet dataset.

### 4.2.1 ImageNet

The quantitative results for this experiment are summarized in Table 3. Similarly to previous experiments, our method is very competitive across all metrics. It should be noted that the quantitative results for DPS-t and DPS have not been reported in Table 3 because DPS methods led to explosion gradients for most of the tested images considered. Figure 4 depicts three examples of a binomial denoising experiment where we observe that our method successfully recovers fine details that PnP-BM3D is not able to capture. In Figure 5, we show another three example cases of a binomial denoising experiment with $\alpha = 2.5, t = 10$, where DPS-t did not lead to exploding gradients. We observe that the recoveries of DPS include finer details than PnP-BM3D but they are also noisy even after finetuning. For instance, using the same hyperparameter

settings in DPS that yield a smooth reconstruction in the third row of Figure 5 will result in noisy recoveries in the other two rows. In this experiment, our method requires 93 seconds per image on average, whereas DPS-t and DPS need 520 seconds per image. See Table 5.

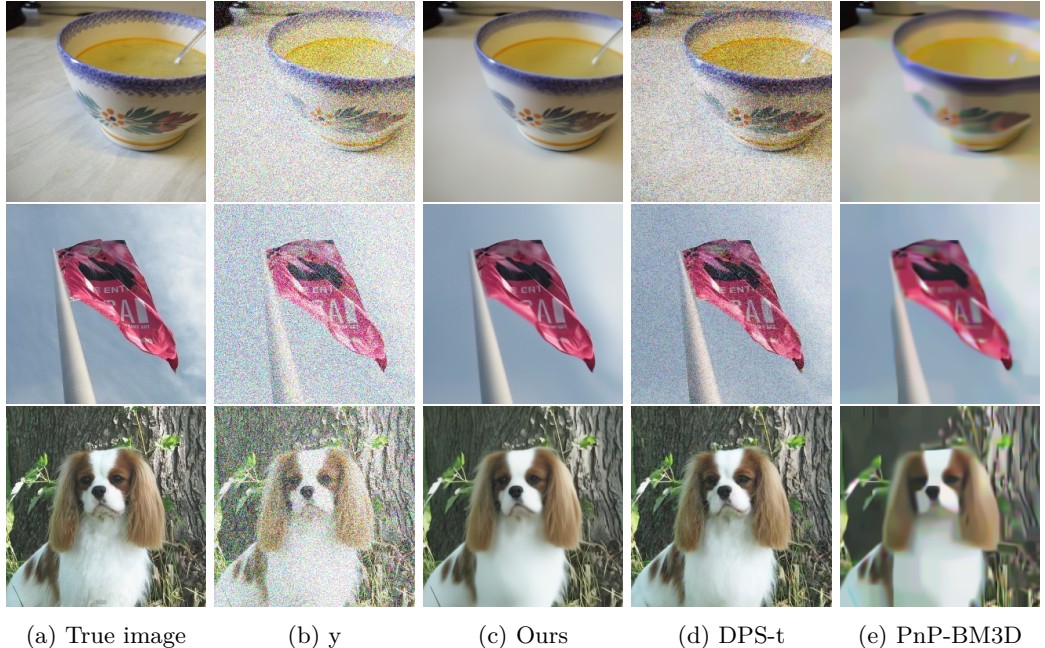

(a) True image       (b) y       (c) Ours       (d) DPS-t       (e) PnP-BM3D

Figure 5: Binomial denoising example ($\alpha = 2.5, t = 10$) on ImageNet dataset.

### 4.2.2 FFHQ

The quantitative results for this dataset are summarized in Table 3. Again, we observe that our method is very competitive in terms of all metrics considered (outperforming competing strategies significantly in PSNR and LPIPS). It should be noted that quantitative results for DPS-t and DPS have not been reported in Table 3 because DPS methods led to exploding gradients for all the testing images. In Figure 6, we illustrate two examples of a binomial denoising problem in the case of $\alpha = 2.5, t = 10$. Note that our method delivers sharp restorations that clearly outperform PnP-BM3D. Our method requires 80 seconds per image, while DPS-t and DPS require 110 seconds. See Table 6.

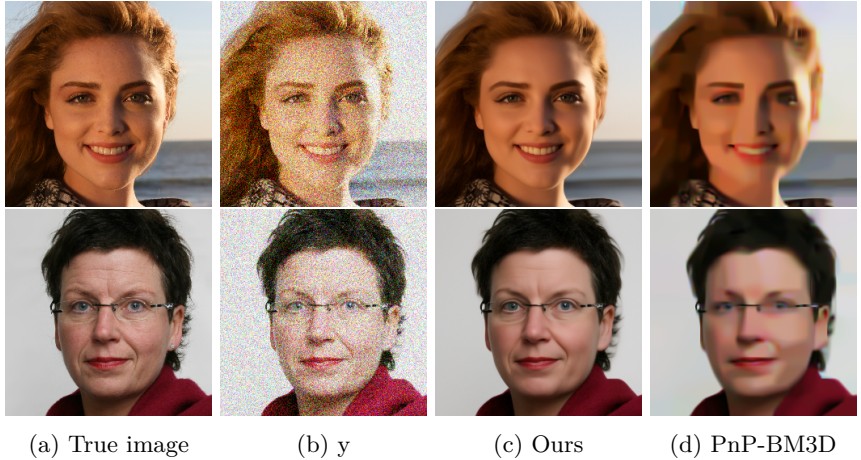

(a) True image       (b) y       (c) Ours       (d) PnP-BM3D

Figure 6: Binomial denoising example ($\alpha = 2.5, t = 10$) on FFHQ dataset.

### 4.3 Geometric inpainting

Lastly, we now consider an extremely challenging geometric inpainting problem, where we seek to estimate $x^\star$ from a measurement $y \sim \mathcal{G}eo(1-e^{-\mathcal{H}(x^\star)})$ where $\mathcal{H}$ is a random mask with Bernoulli random entries with probability $p = 0.5$. We report results for the ImageNet and FFHQ datasets, as summarized below.

Table 4: Geometric inpainting experiment: quantitative results for the FFHQ and ImageNet validation datasets (30 test images per dataset). The best result is shown in bold and the second best is underlined.

| Noise Level | Dataset | Metrics | Ours | PnP-BM3D |
|---|---|---|---|---|
| $\alpha = 0.025$ | FFHQ | PSNR | **22.67** | 21.52 |
| | | SSIM | **0.72** | 0.61 |
| | | LPIPS | **0.31** | 0.55 |
| | ImageNet | PSNR | **21.03** | 20.50 |
| | | SSIM | **0.61** | 0.57 |
| | | LPIPS | **0.44** | 0.54 |

### 4.3.1 ImageNet

The quantitative results for this last experiment are summarized in Table 4. Three representative examples are presented in Figure 7. As we can see in Table 4, our method outperforms PnP-BM3D in all metrics. PnP-BM3D is competitive in PSNR and SSIM since it succeeds in removing heavy noise, but it is less competitive in LPIPS since it cannot recover fine details. It is important to highlight that the original DPS method proposed by Chung et al. (2023) encountered exploding gradients. Figure 7 shows that our method provides accurate reconstructions. Note that the DPS method is not applicable due to exploding gradients. In this experiment, our method requires 260 seconds per image on average. See Table 5.

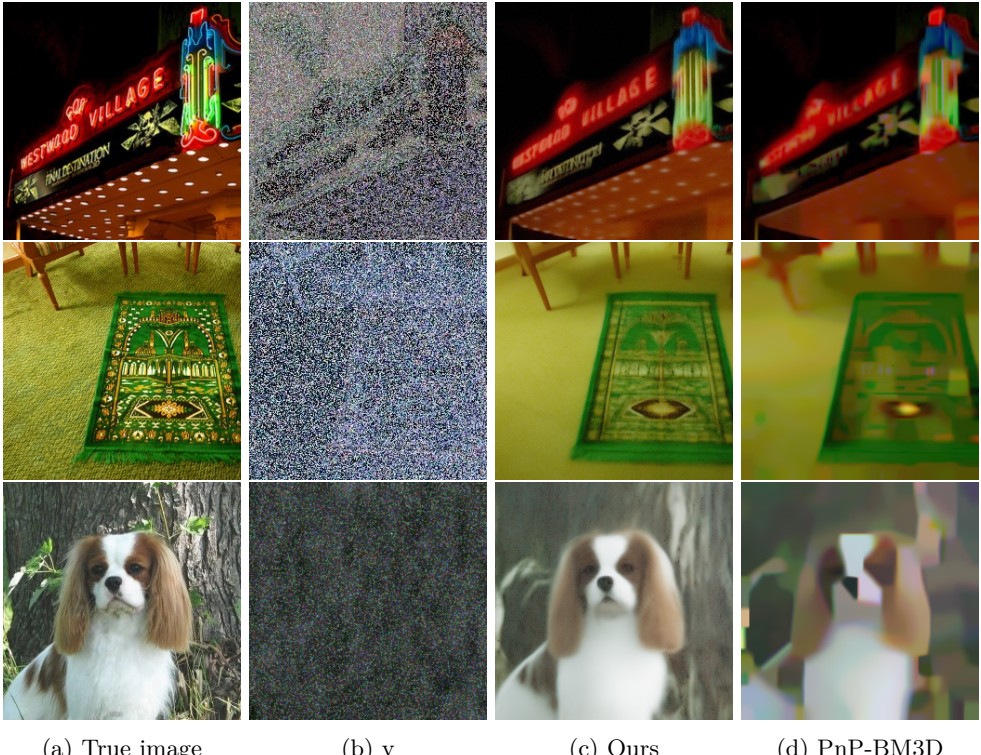

(a) True image      (b) y      (c) Ours      (d) PnP-BM3D

Figure 7: Geometric inpainting example ($\alpha = 0.025$) on ImageNet dataset.

### 4.3.2 FFHQ

The quantitative results for this dataset are summarized in Table 4. The proposed method clearly outperforms PnP-BM3D as a competing strategy, achieving significantly better PSNR and LPIPS values. It is important to highlight that the original DPS method proposed by Chung et al. (2023) encountered exploding gradients. In Figure 8, we present three representative examples of the challenging inpainting task considered. We observe that the proposed method delivers accurate estimates, whereas PnP-BM3D fails to recover fine detail. In this experiment, our method requires 215 seconds per image on average. See Table 6.

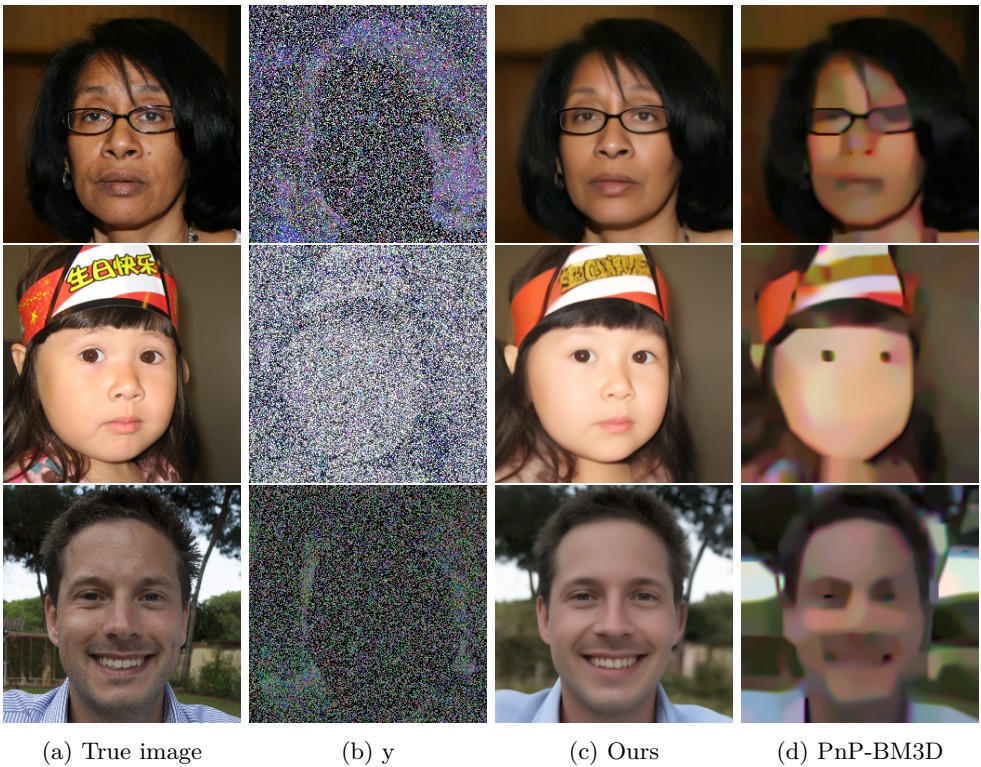

(a) True image      (b) y      (c) Ours      (d) PnP-BM3D

Figure 8: Geometric inpainting example ($\alpha = 0.025$) on FFHQ dataset.

## 5 Conclusion and limitations

This paper presents ProxDiffPIR, a generalisation of the score-based denoising diffusion model DiffPIR for PnP image restoration problems involving photon-starved measurements. The proposed method can handle any likelihood function that is log-concave and potentially non-smooth. In particular, this allows the application of the method to problems involving Poisson, binomial, or geometric likelihood functions, as commonly encountered in modern quantum-enhanced imaging problems. The method combines a foundational pre-trained diffusion model as an implicit prior and a likelihood function specified during test time. The likelihood is involved through a penalised least-squares step that we solve iteratively by using a quasi-Newton method. Extensive image restoration experiments in challenging scenarios with binomial, geometric, or low-intensity Poisson noise show that our method consistently delivers accurate solutions and outperforms alternative strategies from the state-of-the-art.

The computational time of our proposed method ProxDiffPIR and the baseline DPS are summarized in Table 5 and Table 6. As we can see the computational efficiency of ProxDiffPIR is directly linked to the iterative solution of the penalised least-squares step. As shown in the tables, computational time increases with the complexity of the problems, progressing from Poisson noise to binomial and geometric noise. Despite this, ProxDiffPIR is consistently faster than DPS in all cases, except for geometric inpainting on the FFHQ

dataset. However, as indicated in Table 4, DPS suffers from explosive gradients, leading to failed image reconstructions. Future work should explore more efficient solvers for specific image restoration problems of interest. Moreover, although this correction step promotes measurement consistency, there remains some bias. An interesting direction for future research would be to explore randomised correction steps that reduce this bias. Also, although the reconstruction quality of our approach is superior to all other methods, it would be interesting to develop other techniques that are better able to recover subtle details.

Lastly, in Appendix C, we explore several modifications of the original DPS methods that can possibly lead to more stable algorithms. Further results based on these modifications are given in Appendix D. For some problems and experiments, our proposed modifications to DPS are able to improve its stability and deliver restorations that are comparable to ProxDiffPIR. However, modified DPS remains significantly slower than ProxDiffPIR.

Table 5: Computational time comparison of ProxDiffPIR and DPS for experiments conducted on the ImageNet dataset.

| ImageNet dataset | ProxDiffPir | DPS |
|---|---|---|
| Poisson | **37** (s/per image) | 520 (s/per image) |
| Binomial | **93** (s/per image) | 520 (s/per image) |
| Geometric | **260** (s/per image) | 520 (s/per image) |

Table 6: Computational time comparison of ProxDiffPIR and DPS for experiments conducted on the FFHQ dataset.

| FFHQ dataset | ProxDiffPir | DPS |
|---|---|---|
| Poisson | **23** (s/per image) | 110 (s/per image) |
| Binomial | **80** (s/per image) | 110 (s/per image) |
| Geometric | 210 (s/per image) | **110** (s/per image) |

**Acknowledgments**

This work was supported by the UK Research and Innovation (UKRI) Engineering and Physical Sciences Research Council (EPSRC) through grants EP/T007346/1, EP/V006134/1, EP/V006177/1, and EP/W007673/1, and by the EPSRC Centre for Doctoral Training in Mathematical Modelling, Analysis, and Computation (MAC-MIGS), funded by the EPSRC (grant EP/S023291/1), Heriot-Watt University, and the University of Edinburgh. SM acknowledges that the research contribution on this work was undertaken while he was affiliated with Heriot-Watt University; he has since joined Forschungszentrum Juelich.

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

## A  Hyperparameters Values (ProxDiffPIR)

In Tables 7 and 8 below, we present the hyperparameter values that have been used for the ProxDiffPIR implementation for the low-photon imaging experiments and datasets that have been discussed in the main manuscript. To select the values for $\lambda$ and $\zeta$ that lead to the best performance, a grid search was implemented. Specifically, for all the experiments, the ranges of the hyperparameters were selected as $\lambda \in [1, 10]$ and $\zeta \in [0.1, 0.9]$. The incremental step for $\lambda$ was set to be 1 and for $\zeta$ to be 0.1. The search was implemented by selecting 5 random images from the FFHQ and ImageNet datasets respectively.

Table 7: Hyperparameter selection for ProxDiffPIR for the FFHQ dataset.

| | Poisson (Gaussian blur) | | | Poisson (Motion blur) | | | Bin. Denoising | | Geom. Inpainting |
| | $\alpha = 5$ | $\alpha = 10$ | $\alpha = 20$ | $\alpha = 5$ | $\alpha = 10$ | $\alpha = 20$ | $\alpha = 2.5, t = 10$ | $\alpha = 0.25, t = 10^2$ | $\alpha = 0.025$ |
|---|---|---|---|---|---|---|---|---|---|
| $\zeta$ | 0.5 | 0.4 | 0.4 | 0.5 | 0.2 | 0.6 | 0.9 | 0.9 | 0.7 |
| $\lambda$ | 3 | 5 | 3 | 3 | 7 | 3 | 3 | 3 | 2 |

Table 8: Hyperparameter selection for ProxDiffPIR for the ImageNet dataset.

| | Poisson (Gaussian Blur) | | | Poisson (Motion Blur) | | | Bin. Denoising | | Geom. Inpainting |
| | $\alpha = 5$ | $\alpha = 10$ | $\alpha = 20$ | $\alpha = 5$ | $\alpha = 10$ | $\alpha = 20$ | $\alpha = 2.5, t = 10$ | $\alpha = 0.25, t = 10^2$ | $\alpha = 0.025$ |
|---|---|---|---|---|---|---|---|---|---|
| $\zeta$ | 0.4 | 0.4 | 0.4 | 0.5 | 0.6 | 0.5 | 0.9 | 0.95 | 0.73 |
| $\lambda$ | 5 | 6 | 7 | 6 | 5 | 7 | 3 | 3 | 3 |

# B    Implementation Details for DPS

In this section, we explain in more detail how the state-of-the-art diffusion method DPS (Chung et al., 2023) was implemented to make a fair comparison with ProxDiffPIR. It should be reminded that we implemented two versions of DPS; DPS with a Gaussian likelihood approximation, as originally proposed by the authors in Chung et al. (2023), and DPS as implemented with the true likelihood function $p_0(y|x)$ (DPS-t). It should be noted here that DPS and DPS-t proved to show instabilities in some of the experiments, especially the ones related to binomial and geometric noise. We surmise that a primary cause of these instabilities is that DPS and DPS-t fail to account for the poor regularity properties of the low-photon likelihood terms, which for example are undefined at 0 due to the presence of logarithmic terms.

## B.1    DPS

### B.1.1    Poisson noise

The authors in (Chung et al., 2023) approximate the Poisson likelihood

$$p_0(y|x) = \prod_{j=1}^{n} \frac{(\alpha \cdot [\mathcal{H}(\mathbf{x})]_j)^{\mathbf{y}_j} \exp(-\alpha \cdot [\mathcal{H}(\mathbf{x})]_j)}{\mathbf{y}_j!} \quad , \tag{14}$$

by

$$p_0(y|x) \approx \prod_{j=1}^{n} \frac{1}{\sqrt{2\pi\mathbf{y}_j}} \exp\left(-\frac{(\mathbf{y}_j - [\alpha \cdot \mathcal{H}(\mathbf{x})]_j)^2}{2\mathbf{y}_j}\right) \quad . \tag{15}$$

We present DPS (Chung et al., 2023) in Algorithm 3 for clarity of presentation in the case of Poisson measurements.

---

**Algorithm 3** DPS Chung et al. (2023)

---

**Require:** $s_\theta, T, y, \{\rho_t\}_{t=1}^T, \{\tilde{\sigma}_t\}_{t=1}^T, [\mathbf{\Lambda}]_{jj} = \frac{1}{2\mathbf{y}_j}$

1: $X_T \sim N(0,1)$
2: **for** $t = T - 1 : 0$ **do**
3:     $x_0^{(t)} = \frac{1}{\sqrt{\bar{\alpha}_t}} (x_t + (1 - \bar{\alpha}_t) s_\theta (x_t, t))$
4:     $z \sim N(0,1)$
5:     $x'_{t-1} = \frac{\sqrt{\bar{\alpha}_{t-1}}\beta_t}{1-\bar{\alpha}_t}\hat{x}_0 + \frac{\sqrt{\alpha_t}(1-\bar{\alpha}_{t-1})}{1-\bar{\alpha}_t}x_t + \tilde{\sigma}_t z$
6:     $x_{i-1} = x'_{t-1} - \rho_t \nabla_{x_t} \|y - \mathcal{H}(\hat{x}_0)\|_\Lambda^2$
7: **end for**
8: **Return** $\hat{x}_0$

---

### B.1.2    Binomial noise.

The authors in Chung et al. (2023) do not discuss the case of binomial distributed measurements. In cases where the number of repetition periods is large ($t \to \infty$), the model can be approximated by a Gaussian distribution as

$$p_0(y|x) \approx q_y(x) \quad , \tag{16}$$

where

$$q_y(x) = \prod_{j=1}^{n} \frac{1}{\sqrt{2\pi[\mathbf{tp(1-p)}]_j}} \exp\left(-\frac{(\mathbf{y}_j - [\mathbf{tp}]_j)^2}{2[\mathbf{tp(1-p)}]_j}\right) \quad , \tag{17}$$

where we recall that $\mathbf{p} = 1 - e^{-\alpha \cdot \mathcal{H}(\mathbf{x})}$.

### B.1.3 Finetuning.

The stepsize $\rho_t$ in Algorithm 3 is a crucial hyperparameter for the stability and performance of stabilized DPS. The authors in Chung et al. (2023) set $\rho_t$ as

$$\rho_t = \zeta / \|y - \mathcal{H}(\hat{x}_0(x_t))\| \quad , \tag{18}$$

where $\zeta > 0$ is set as constant. We finetune $\zeta$ by using a grid search. In Tables 9 and 10 we show how $\zeta$ is selected for the several Poisson experiments discussed in the main manuscript for the FFHQ and ImageNet datasets respectively.

Table 9: Hyperparameter selection for DPS for the FFHQ dataset.

|  | Poisson (Gaussian blur) | | | Poisson (Motion blur) | | |
|---|---|---|---|---|---|---|
|  | $\alpha = 5$ | $\alpha = 10$ | $\alpha = 20$ | $\alpha = 5$ | $\alpha = 10$ | $\alpha = 20$ |
| $\zeta$ | 0.5 | 0.008 | 0.01 | 0.6 | 0.005 | 0.012 |

Table 10: Hyperparameter selection for DPS for the ImageNet dataset.

|  | Poisson (Gaussian Blur) | | | Poisson (Motion Blur) | | |
|---|---|---|---|---|---|---|
|  | $\alpha = 5$ | $\alpha = 10$ | $\alpha = 20$ | $\alpha = 5$ | $\alpha = 10$ | $\alpha = 20$ |
| $\zeta$ | 0.2 | 0.008 | 0.008 | 0.4 | 0.003 | 0.003 |

## B.2 DPS-t

The DPS algorithm can be implemented with the true likelihood function $p_0(y|x)$. For clarity, we refer to this version as DPS-t. For the Poisson, binomial and geometric log-likelihoods $f_y$, as presented in the main manuscript, we can implement DPS-t as shown in Algorithm 4.

---

**Algorithm 4** DPS-t

---

**Require:** $s_\theta, T, y, \{\zeta_t\}_{t=1}^{T}, \{\tilde{\sigma}_t\}_{t=1}^{T}$
1: $X_T \sim N(0,1)$
2: **for** $t = T-1 : 0$ **do**
3: $\quad x_0^{(t)} = \frac{1}{\sqrt{\bar{\alpha}_t}}(x_t + (1 - \bar{\alpha}_t) s_\theta(x_t, t))$
4: $\quad z \sim N(0,1)$
5: $\quad x'_{t-1} = \frac{\sqrt{\bar{\alpha}_{t-1}}\beta_t}{1-\bar{\alpha}_t}\hat{x}_0 + \frac{\sqrt{\alpha_t}(1-\bar{\alpha}_{t-1})}{1-\bar{\alpha}_t}x_t + \tilde{\sigma}_t z$
6: $\quad x_{i-1} = x'_{t-1} - \zeta \nabla_{x_t} f(y, \mathcal{H}(\hat{x}_0))$
7: **end for**
8: **Return** $\hat{x}_0$

---

### B.2.1 Finetuning.

Similarly with DPS, a stepsize needs to be included in DPS-t. We assume a constant stepsize $\zeta$, see Algorithm 4. We finetune $\zeta$ by implementing a grid search on 5 random images from the FFHQ and ImageNet datasets. In Tables 11 and 12 we list how the hyperparameter $\zeta$ is selected for the several experiments discussed in the main manuscript for the FFHQ and ImageNet datasets, respectively.

Table 11: Hyperparameter selection for DPS-t for the FFHQ dataset.

| | Poisson (Gaussian blur) | | | Poisson (Motion blur) | | |
|---|---|---|---|---|---|---|
| | $\alpha = 5$ | $\alpha = 10$ | $\alpha = 20$ | $\alpha = 5$ | $\alpha = 10$ | $\alpha = 20$ |
| $\zeta$ | 400 | 450 | 300 | 200 | 250 | 200 |

Table 12: Hyperparameter selection for DPS-t for the ImageNet dataset.

| | Poisson (Gaussian Blur) | | | Poisson (Motion Blur) | | |
|---|---|---|---|---|---|---|
| | $\alpha = 5$ | $\alpha = 10$ | $\alpha = 20$ | $\alpha = 5$ | $\alpha = 10$ | $\alpha = 20$ |
| $\zeta$ | 450 | 400 | 150 | 450 | 450 | 300 |

## C   Modifications on DPS and DPS-t

In this section, we propose stabilized versions of DPS and DPS-t that deal with the poor properties, providing more stable results than their original versions depicted in Chung et al. (2023).

### C.1   Stabilized DPS

Regarding DPS, we simply propose to approximate the variance term as

$$[\mathbf{\Lambda}^\beta]_{jj} = \frac{1}{2[\mathbf{y}]_j + \beta} \quad , \tag{19}$$

which leads to the approximation

$$p_0(y|x) \approx \prod_{j=1}^n \frac{1}{\sqrt{2\pi \mathbf{y}_j}} \exp\left(-\frac{(\mathbf{y}_j - [\alpha \cdot \mathcal{H}(\mathbf{x})]_j)^2}{2([\mathbf{y}]_j + \beta)}\right) \quad , \tag{20}$$

where $\beta > 0$ takes a small value compared to the scale of $\alpha \cdot x$. This guarantees that no instabilities will occur when $y = 0$. We show the stabilized DPS for Poisson measurements in Algorithm 5.

In the case of binomial noise, we also observed in our experiments that DPS was inapplicable leading to exploding gradients. DPS can be modified for binomial-distributed measurements to give more stable results similarly with the Poisson setting by approximating $\mathbf{p} = 1 - e^{-\alpha \cdot \mathcal{H}(\mathbf{x})}$ by $\mathbf{p}^\beta = 1 - e^{-(\alpha \cdot \mathcal{H}(\mathbf{x}) + \beta)}$, for $\beta > 0$. See Algorithm 6 for more details and Tables 13 and 14 for the specific values of $\beta$ used in further experiments in Appendix D.

---

**Algorithm 5** Stabilized DPS (Poisson measurements)

---

**Require:** $s_\theta, T, y, \{\rho_t\}_{t=1}^T, \{\tilde{\sigma}_t\}_{t=1}^T, \beta, [\mathbf{\Lambda}^\beta]_{jj} = \frac{1}{2\mathbf{y}_j + \beta}$

1: $X_T \sim N(0, 1)$
2: **for** $t = T - 1 : 0$ **do**
3: $\quad x_0^{(t)} = \frac{1}{\sqrt{\bar{\alpha}_t}} \left(x_t + (1 - \bar{\alpha}_t) s_\theta(x_t, t)\right)$
4: $\quad z \sim N(0, 1)$
5: $\quad x'_{t-1} = \frac{\sqrt{\bar{\alpha}_{t-1}}\beta_t}{1 - \bar{\alpha}_t}\hat{x}_0 + \frac{\sqrt{\alpha_t}(1 - \bar{\alpha}_{t-1})}{1 - \bar{\alpha}_t}x_t + \tilde{\sigma}_t z$
6: $\quad x_{i-1} = x'_{t-1} - \rho_t \nabla_{x_t} \|y - \mathcal{H}(\hat{x}_0)\|_{\mathbf{\Lambda}^\beta}^2$
7: **end for**
8: **Return** $\hat{x}_0$

---

### C.1.1   Finetuning.

The stepsize $\rho_t$ in Algorithm 5 is a crucial hyperparameter for the stability and performance of stabilized DPS. The authors in Chung et al. (2023) set $\rho_t$ as

$$\rho_t = \zeta / \|y - \mathcal{H}(\hat{x}_0(x_t))\| \quad , \tag{21}$$

---

**Algorithm 6** Stabilized DPS (binomial measurements)

---

**Require:** $s_\theta, T, y, \{\zeta_t\}_{t=1}^T, \{\tilde{\sigma}_t\}_{t=1}^T$

1: $X_T \sim N(0, 1)$
2: **for** $t = T - 1 : 0$ **do**
3:      $x_0^{(t)} = \frac{1}{\sqrt{\bar{\alpha}_t}} (x_t + (1 - \bar{\alpha}_t) s_\theta (x_t, t))$
4:      $z \sim N(0, 1)$
5:      $x'_{t-1} = \frac{\sqrt{\bar{\alpha}_{t-1}} \beta_t}{1 - \bar{\alpha}_t} \hat{x}_0 + \frac{\sqrt{\alpha_t}(1 - \bar{\alpha}_{t-1})}{1 - \bar{\alpha}_t} x_t + \tilde{\sigma}_t z$
6:      $x_{i-1} = x'_{t-1} - \zeta \nabla_{x_t} q_y(\hat{x}_0)$
7: **end for**
8: **Return** $\hat{x}_0$

---

where $\zeta > 0$ is set as constant. We finetune $\zeta$ by using a grid search. In Tables 13 and 14 we show how $\zeta$ is selected for the several Poisson experiments discussed in the main manuscript for the FFHQ and ImageNet datasets respectively.

Table 13: Hyperparameter selection for stabilized DPS for the FFHQ dataset.

|  | Poisson (Gaussian blur) | | | Poisson (Motion blur) | | | Bin. Denoising | |
| --- | --- | --- | --- | --- | --- | --- | --- | --- |
|  | $\alpha = 5$ | $\alpha = 10$ | $\alpha = 20$ | $\alpha = 5$ | $\alpha = 10$ | $\alpha = 20$ | $\alpha = 2.5, t = 10$ | $\alpha = 0.25, t = 10^2$ |
| $\zeta$ | 0.4 | 0.6 | 1 | 0.5 | 0.6 | 0.9 | 0.0015 | 0.0006 |
| $\beta$ | 0.1 | 0.1 | 0.1 | 0.1 | 0.1 | 0.1 | 0.1 | 0.01 |

Table 14: Hyperparameter selection for stabilized DPS for the ImageNet dataset.

|  | Poisson (Gaussian blur) | | | Poisson (Motion blur) | | | Bin. Denoising | |
| --- | --- | --- | --- | --- | --- | --- | --- | --- |
|  | $\alpha = 5$ | $\alpha = 10$ | $\alpha = 20$ | $\alpha = 5$ | $\alpha = 10$ | $\alpha = 20$ | $\alpha = 2.5, t = 10$ | $\alpha = 0.25, t = 10^2$ |
| $\zeta$ | 0.14 | 0.12 | 0.16 | 0.1 | 0.1 | 0.1 | 0.0006 | 0.0009 |
| $\beta$ | 0.1 | 0.1 | 0.1 | 0.1 | 0.1 | 0.1 | 0.1 | 0.01 |

## C.2 Stabilized DPS-t

As we have observed in Section 4, DPS-t was either inapplicable or led to suboptimal results . Note that we can approximate $f_y$ by its stabilized version $f_y^\beta$, similar to ProxDiffPIR, see Algorithm 7. We noticed that this can lead to more stable samples and results.

---

**Algorithm 7** Stabilized DPS-t

---

**Require:** $s_\theta, T, y, \{\zeta_t\}_{t=1}^T, \{\tilde{\sigma}_t\}_{t=1}^T, \beta$

1: $X_T \sim N(0, 1)$
2: **for** $t = T - 1 : 0$ **do**
3:      $x_0^{(t)} = \frac{1}{\sqrt{\bar{\alpha}_t}} (x_t + (1 - \bar{\alpha}_t) s_\theta (x_t, t))$
4:      $z \sim N(0, 1)$
5:      $x'_{t-1} = \frac{\sqrt{\bar{\alpha}_{t-1}} \beta_t}{1 - \bar{\alpha}_t} \hat{x}_0 + \frac{\sqrt{\alpha_t}(1 - \bar{\alpha}_{t-1})}{1 - \bar{\alpha}_t} x_t + \tilde{\sigma}_t z$
6:      $x_{i-1} = x'_{t-1} - \zeta \nabla_{x_t} f(y, \mathcal{H}(\hat{x}_0) + \beta)$
7: **end for**
8: **Return** $\hat{x}_0$

---

### C.2.1 Finetuning

Similarly with stabilized DPS, a stepsize needs to be included in stabilized DPS-t. We assume a constant stepsize $\zeta$, see Algorithm 7. We finetune $\zeta$ by implementing a grid search on 5 random images from the FFHQ and ImageNet datasets. In Tables 15 and 16 we list how the hyperparameters $\zeta$ and $\beta$ are selected for the several experiments discussed in the main manuscript for the FFHQ and ImageNet datasets, respectively.

Table 15: Hyperparameter selection for stabilized DPS-t for the FFHQ dataset.

| | Poisson (Gaussian blur) | | | Poisson (Motion blur) | | | Bin. Denoising | | Geom. Inpainting |
| | $\alpha = 5$ | $\alpha = 10$ | $\alpha = 20$ | $\alpha = 5$ | $\alpha = 10$ | $\alpha = 20$ | $\alpha = 2.5, t = 10$ | $\alpha = 0.25, t = 10^2$ | $\alpha = 0.025$ |
|---|---|---|---|---|---|---|---|---|---|
| $\zeta$ | 300 | 250 | 200 | 520 | 400 | 300 | 180 | 150 | 400 |
| $\beta$ | 0.025 | 0.05 | 0.1 | 0.025 | 0.05 | 0.1 | 0.01 | 0.001 | 0.001 |

Table 16: Hyperparameter selection for stabilized DPS-t for the ImageNet dataset.

| | Poisson (Gaussian blur) | | | Poisson (Motion blur) | | | Bin. Denoising | | Geom. Inpainting |
| | $\alpha = 5$ | $\alpha = 10$ | $\alpha = 20$ | $\alpha = 5$ | $\alpha = 10$ | $\alpha = 20$ | $\alpha = 2.5, t = 10$ | $\alpha = 0.25, t = 10^2$ | $\alpha = 0.025$ |
|---|---|---|---|---|---|---|---|---|---|
| $\zeta$ | 220 | 100 | 200 | 300 | 200 | 150 | 220 | 200 | 500 |
| $\beta$ | 0.025 | 0.05 | 0.05 | 0.025 | 0.05 | 0.1 | 0.01 | 0.001 | 0.001 |

# D   Further Results

## D.1   Poisson noise

In Tables 17 and 18, we show the quantitative comparison between ProxDiffPIR, stabilized DPS, DPS, stabilized DPS-t and DPS-t for the FFHQ and ImageNet datasets, respectively. For the FFHQ dataset, among all DPS versions, we observe that stabilized DPS-t is the most competitive in terms of all metrics. Importantly, in cases where DPS-t is not applicable (see, for example, the cases where a motion blur has been used in the inverse problem), stabilized DPS-t is not only applicable, but also improves significantly over DPS-t; see also Figure 9-f and Figure 9-g for a visual comparison. In Figure 10, we also observe that stabilized DPS versions can correct some bias, but that might not always be the case; For example, compare Figure 9-d and Figure 9-e where the recovery from stabilized DPS does not improve over the recovery of DPS. Regarding ProxDiffPIR, we observe that it provides very competitive results in terms of PSNR and SSIM while it is slightly less competitive in terms of LPIPS than stabilized DPS-t. For the ImageNet dataset, stabilized DPS-t is not as competitive as stabilized DPS; however, it can provide more competitive results in terms of PSNR in some cases. see Table 18. As illustrated in Figure 11, the stabilized versions of DPS show significant improvements compared to the original DPS versions, which corroborates the quantitative findings detailed in Table 18. However, regardless the finetuning of all DPS versions, instabilities can still occur even by using the stabilized versions of DPS, see Figure 12. Finally, it should be observed in Table 18 that ProxDiffPIR still offers more competitive performance in terms of all metrics than any version of DPS.

Table 17: Quantitative mean results over 30 images from the FFHQ validation dataset for the **Poisson deblurring** problem.

| Noise Level | Kernel | Metrics | ProxDiffPIR | Stabilized DPS | DPS | Stabilized DPS-t | DPS-t |
|---|---|---|---|---|---|---|---|
| $\alpha = 5$ high noise | Gaussian blur | PSNR | **24.77** | 23.57 | 22.55 | 23.48 | 20.35 |
| | | SSIM | **0.70** | 0.68 | 0.68 | 0.69 | 0.49 |
| | | LPIPS | 0.33 | 0.31 | 0.31 | **0.30** | 0.46 |
| | Motion blur | PSNR | **25.05** | 23.23 | 22.97 | 24.02 | - |
| | | SSIM | **0.72** | 0.64 | 0.70 | 0.70 | - |
| | | LPIPS | 0.31 | 0.35 | **0.30** | **0.30** | - |
| $\alpha = 10$ medium noise | Gaussian blur | PSNR | **25.12** | 23.79 | 23.65 | 24.07 | 19.94 |
| | | SSIM | **0.71** | 0.67 | 0.70 | **0.71** | 0.42 |
| | | LPIPS | 0.31 | 0.32 | **0.29** | **0.29** | 0.52 |
| | Motion blur | PSNR | **25.12** | 23.57 | 23.70 | 24.70 | - |
| | | SSIM | 0.72 | 0.65 | 0.71 | **0.73** | - |
| | | LPIPS | 0.29 | 0.35 | **0.29** | **0.28** | - |
| $\alpha = 20$ low noise | Gaussian blur | PSNR | **25.85** | 23.36 | 24.23 | 24.43 | 20.20 |
| | | SSIM | **0.74** | 0.64 | 0.71 | 0.72 | 0.40 |
| | | LPIPS | 0.29 | 0.35 | **0.28** | **0.28** | 0.52 |
| | Motion blur | PSNR | **26.50** | 22.97 | 24.78 | 25.10 | - |
| | | SSIM | **0.77** | 0.62 | 0.74 | 0.74 | - |
| | | LPIPS | 0.29 | 0.37 | **0.27** | 0.28 | - |

Table 18: Quantitative mean results over 30 images from the ImageNet validation dataset for the **Poisson deblurring** problem.

| Noise Level | Kernel | Metrics | ProxDiffPIR | Stabilized DPS | DPS | Stabilized DPS-t | DPS-t |
|---|---|---|---|---|---|---|---|
| | | PSNR | **22.60** | 21.16 | 20.25 | 20.40 | 16.66 |
| | Gaussian blur | SSIM | **0.60** | 0.56 | 0.55 | 0.52 | 0.26 |
| $\alpha = 5$ | | LPIPS | **0.41** | **0.41** | 0.42 | 0.44 | 0.60 |
| high noise | | PSNR | **22.86** | 21.25 | 21.01 | 21.52 | - |
| | Motion blur | SSIM | **0.61** | 0.57 | 0.60 | 0.59 | - |
| | | LPIPS | 0.41 | 0.41 | **0.39** | **0.39** | - |
| | | PSNR | **23.14** | 20.58 | 17.59 | 20.07 | 16.55 |
| | Gaussian blur | SSIM | **0.63** | 0.51 | 0.30 | 0.52 | 0.24 |
| $\alpha = 10$ | | LPIPS | **0.39** | 0.43 | 0.56 | 0.44 | 0.62 |
| medium noise | | PSNR | **23.75** | 21.68 | 21.56 | 21.72 | - |
| | Motion blur | SSIM | **0.66** | 0.60 | 0.61 | 0.59 | - |
| | | LPIPS | **0.38** | 0.39 | **0.38** | 0.39 | - |
| | | PSNR | **23.63** | 20.16 | 18.15 | 19.47 | 17.87 |
| | Gaussian blur | SSIM | **0.65** | 0.48 | 0.33 | 0.49 | 0.31 |
| $\alpha = 20$ | | LPIPS | **0.37** | 0.45 | 0.54 | 0.47 | 0.57 |
| low noise | | PSNR | **24.16** | 21.83 | 21.64 | 21.52 | - |
| | Motion blur | SSIM | 0.60 | **0.61** | 0.60 | 0.55 | - |
| | | LPIPS | **0.36** | 0.38 | 0.40 | 0.43 | - |

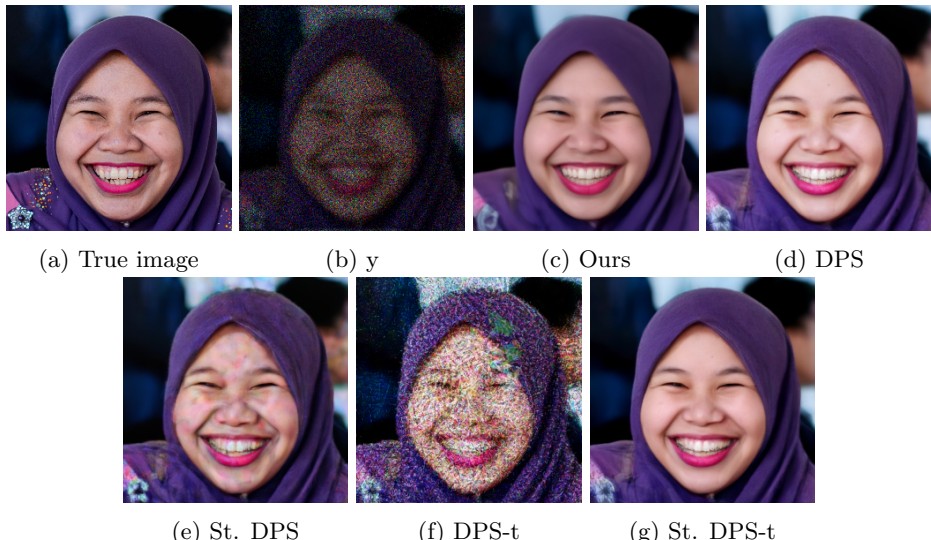

(a) True image      (b) y      (c) Ours      (d) DPS

(e) St. DPS      (f) DPS-t      (g) St. DPS-t

Figure 9: Poisson deblurring example ($\alpha = 5$) with gaussian blur on the FFHQ dataset.

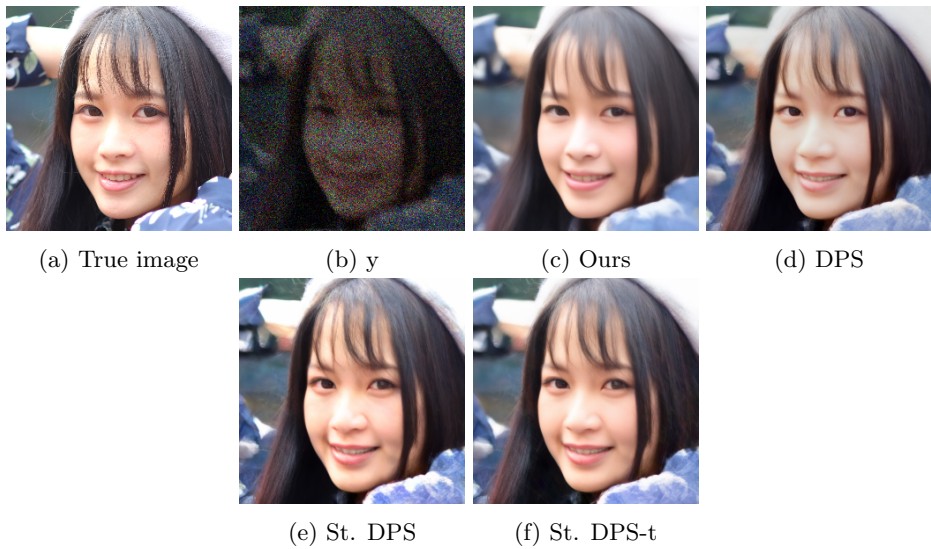

Figure 10: Poisson deblurring example ($\alpha = 5$) with motion blur on the FFHQ dataset.

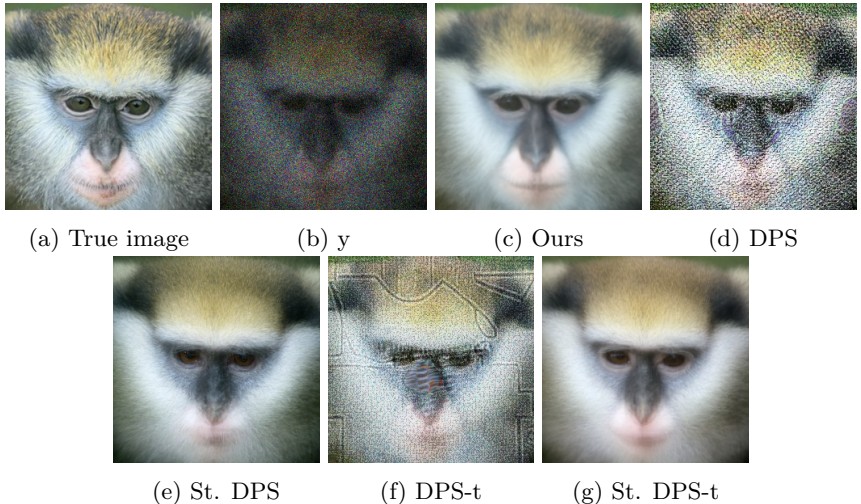

Figure 11: Poisson deblurring example ($\alpha = 10$) with Gaussian blur on ImageNet dataset.

## D.2 Binomial noise

In Table 19, we show the quantitative comparison between ProxDiffPIR, and all versions of DPS for the FFHQ and ImageNet datasets. We observe that for both datasets stabilized DPS and stabilized DPS-t do not lead to exploding gradients as DPS and DPS-t do. We also observe that ProxDiffPIR provides very competitive results in terms of all considered metrics and datasets with stabilized DPS-t being competitive in terms of LPIPS. In Section 4, there were example cases for which the original versions of DPS were inapplicable see Figures 6 and 4. This is not the case now by making use of the stabilized versions of DPS since they can provide more stable results and avoid exploding gradients, see Figures 13 and 14 instead. By comparison to DPS strategies, our method delivers results that are more smooth (see, e.g., the first row of Figure 13), hence DPS-t achieves a mildly better LPIPS value. However, DPS strategies might still deliver solutions that appear realistic but are not in agreement with the truth (e.g., see the colour of the background in the second row of Figure 13 as well as the examples in Figure 14). In Figure 15, we visualize additional comparisons between ProxDiffPIR, stabilized DPS, DPS-t and stabilized DPS-t (original DPS led to exploding gradients for these images). A first observation is how stabilized DPS-t can significantly

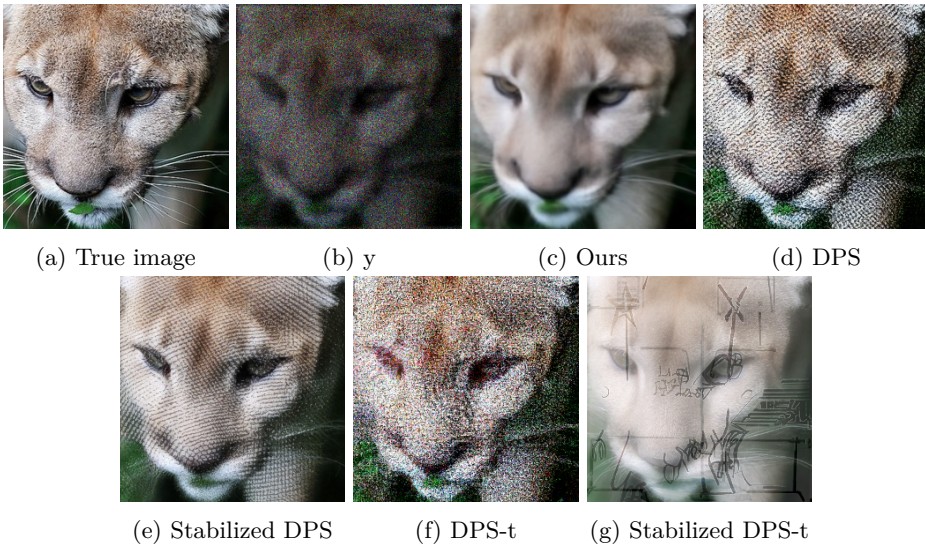

(a) True image      (b) y      (c) Ours      (d) DPS

(e) Stabilized DPS      (f) DPS-t      (g) Stabilized DPS-t

Figure 12: Poisson deblurring example ($\alpha = 10$) with Gaussian blur on ImageNet dataset.

improve over original DPS-t providing more stable, smooth and fine-detailed results. Stabilized DPS can also provide fine-detailed reconstructions but it is prone to artifacts (see bottom row of Figure 15). A second observation is that ProxDiffPIR smooths out some details compared to stabilized DPS-t for the example images. Taking into account Figure 14) though, overall ProxDiffPIR proves to be more stable.

Table 19: Binomial denoising experiment: quantitative image restoration results (averaged over 30 images from the FFHQ and ImageNet validation datasets). For each quality metric, the best result is shown in bold and the second best is underlined.

| Noise Level | Dataset | Metrics | Ours | Stabilized DPS | DPS | Stabilized | DPS-t |
|---|---|---|---|---|---|---|---|
| $\alpha = 2.5, t = 10$ | FFHQ | PSNR | **28.78** | 23.71 | - | 27.25 | - |
| | | SSIM | **0.85** | 0.76 | - | 0.84 | - |
| | | LPIPS | 0.22 | 0.25 | - | **0.21** | - |
| | ImageNet | PSNR | **27.97** | 22.82 | - | 26.85 | - |
| | | SSIM | **0.81** | 0.77 | - | 0.80 | - |
| | | LPIPS | 0.24 | 0.29 | - | **0.24** | - |
| $\alpha = 0.25, t = 100$ | FFHQ | PSNR | **29.83** | 23.32 | - | 28.72 | - |
| | | SSIM | **0.87** | 0.81 | - | **0.87** | - |
| | | LPIPS | 0.20 | 0.22 | - | **0.18** | - |
| | ImageNet | PSNR | **29.45** | 23.70 | - | 29.32 | - |
| | | SSIM | **0.85** | 0.81 | - | **0.85** | - |
| | | LPIPS | 0.20 | 0.21 | - | **0.19** | - |

## D.3 Geometric noise

In Table 20, we show the quantitative comparison between ProxDiffPIR, stabilized DPS-t and DPS-t for the FFHQ and ImageNet datasets. Note that DPS is not used here, as approximating a geometric distribution by using a Gaussian one is, to the best of our knowledge, DPS is not applicable here) invalid. We observe that for both datasets stabilized DPS-t do not lead to exploding gradients as DPS-t does. We also observe that ProxDiffPIR provides very competitive results in terms of all considered metrics and datasets, with stabilized DPS-t being competitive in terms of LPIPS. Although stabilized DPS has the potential to create intricately detailed reconstructions, it may exhibit bias; see the first row of Figure 16 and the first two rows of 17. This assertion is supported by the low PSNR and SSIM performance observed for DPS in Table 20. It should be noted that some bias can be also exhibited by ProxDiffPIR, see for example bottom rows in Figures 16 and 17. However, in our experiments, we did not notice extreme cases of bias as DPS-t exhibits, for example, in the first two rows of 17. This assertion is supported by the higher PSNR and SSIM performance of ProxDiffPIR compared to DPS-t.

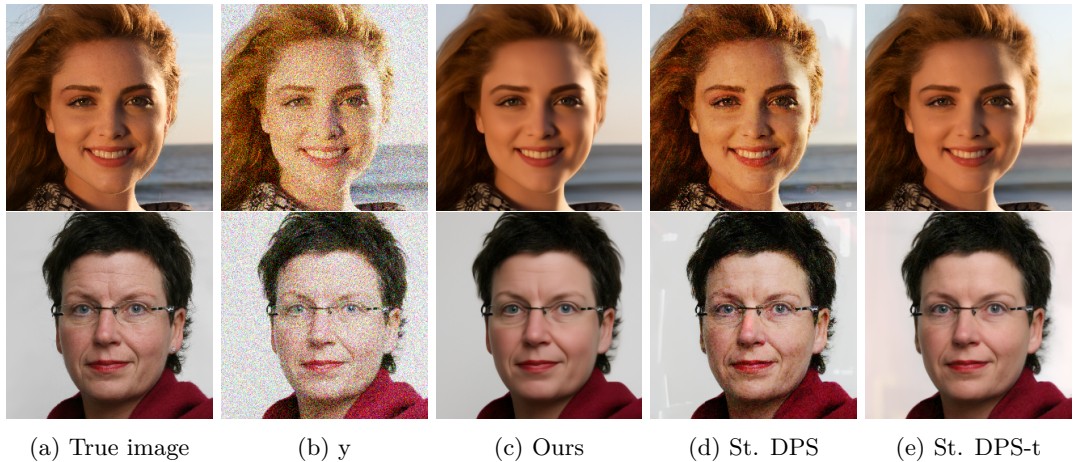

(a) True image      (b) y      (c) Ours      (d) St. DPS      (e) St. DPS-t

Figure 13: Binomial denoising example ($\alpha = 2.5, t = 10$) on FFHQ dataset.

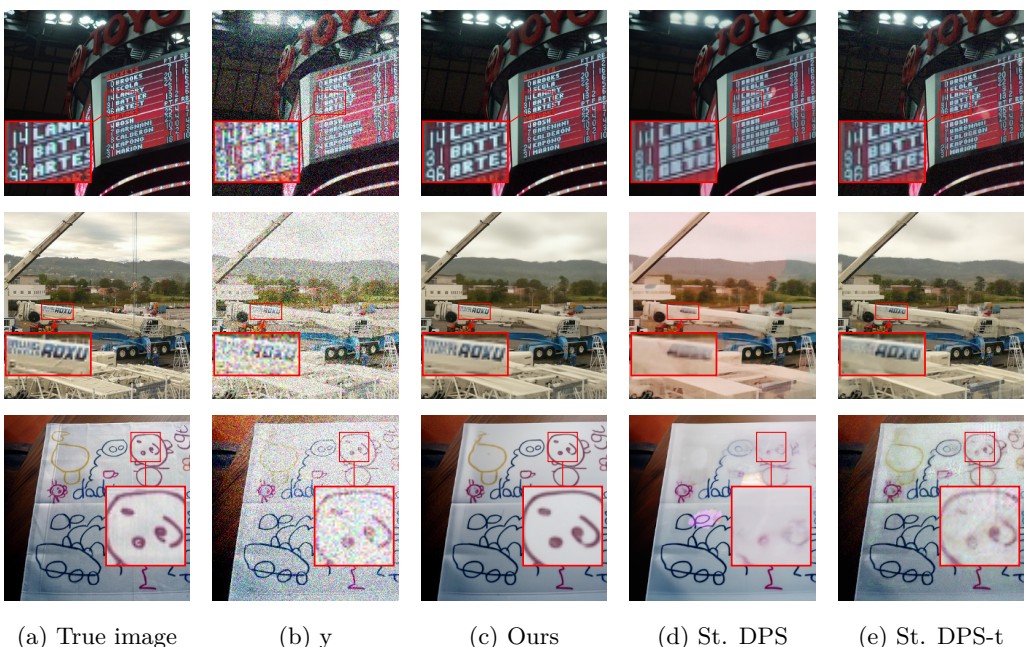

(a) True image      (b) y      (c) Ours      (d) St. DPS      (e) St. DPS-t

Figure 14: Binomial denoising example ($\alpha = 2.5, t = 10$) on ImageNet dataset.

Table 20: Geometric inpainting experiment: quantitative results for the FFHQ and ImageNet validation datasets (30 test images per dataset). The best result is shown in bold and the second best is underlined.

| Noise Level | Dataset | Metrics | Ours | Stabilized DPS-t | DPS-t |
|---|---|---|---|---|---|
| $\alpha = 0.025$ | FFHQ | PSNR | **22.67** | 19.44 | - |
| | | SSIM | **0.72** | 0.63 | - |
| | | LPIPS | **0.31** | 0.33 | - |
| | ImageNet | PSNR | **21.03** | 15.60 | - |
| | | SSIM | **0.61** | 0.49 | - |
| | | LPIPS | **0.44** | 0.46 | - |

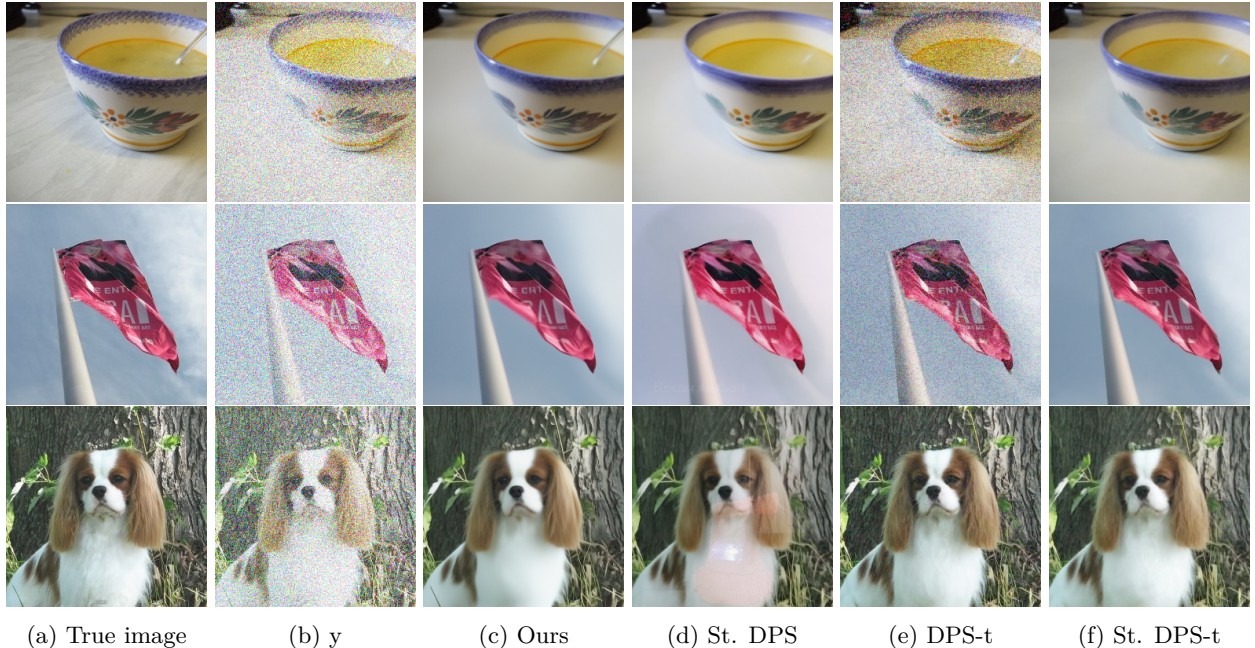

(a) True image    (b) y    (c) Ours    (d) St. DPS    (e) DPS-t    (f) St. DPS-t

Figure 15: Binomial denoising example ($\alpha = 2.5, t = 10$) on ImageNet dataset.

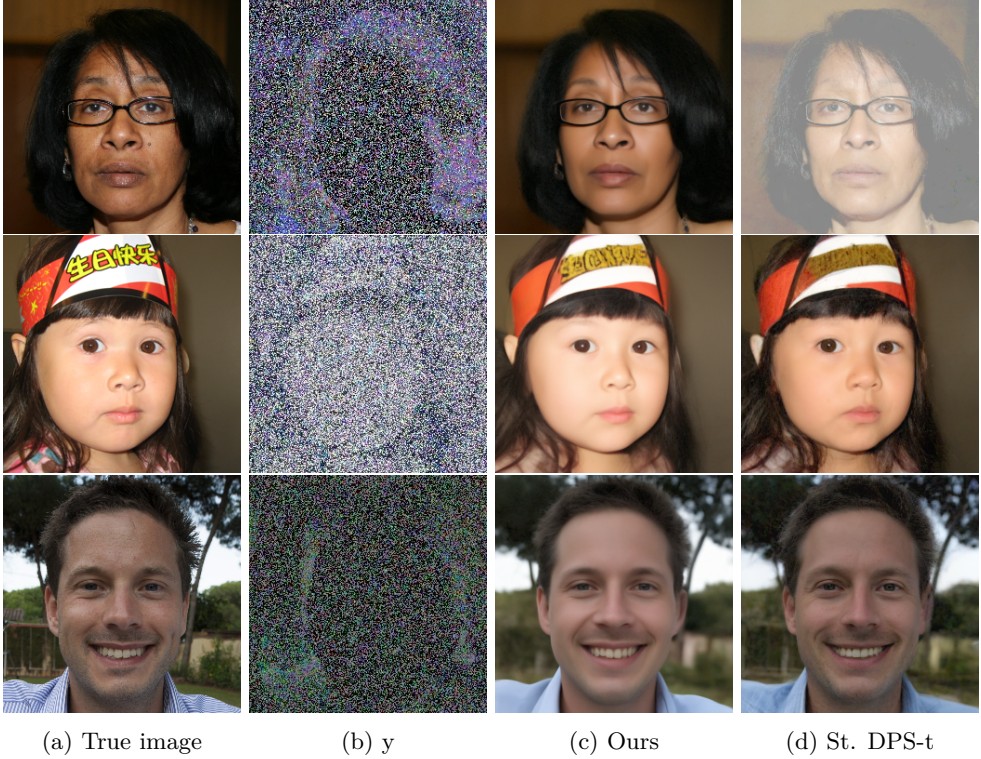

(a) True image    (b) y    (c) Ours    (d) St. DPS-t

Figure 16: Geometric inpainting example ($\alpha = 0.025$) on FFHQ dataset.

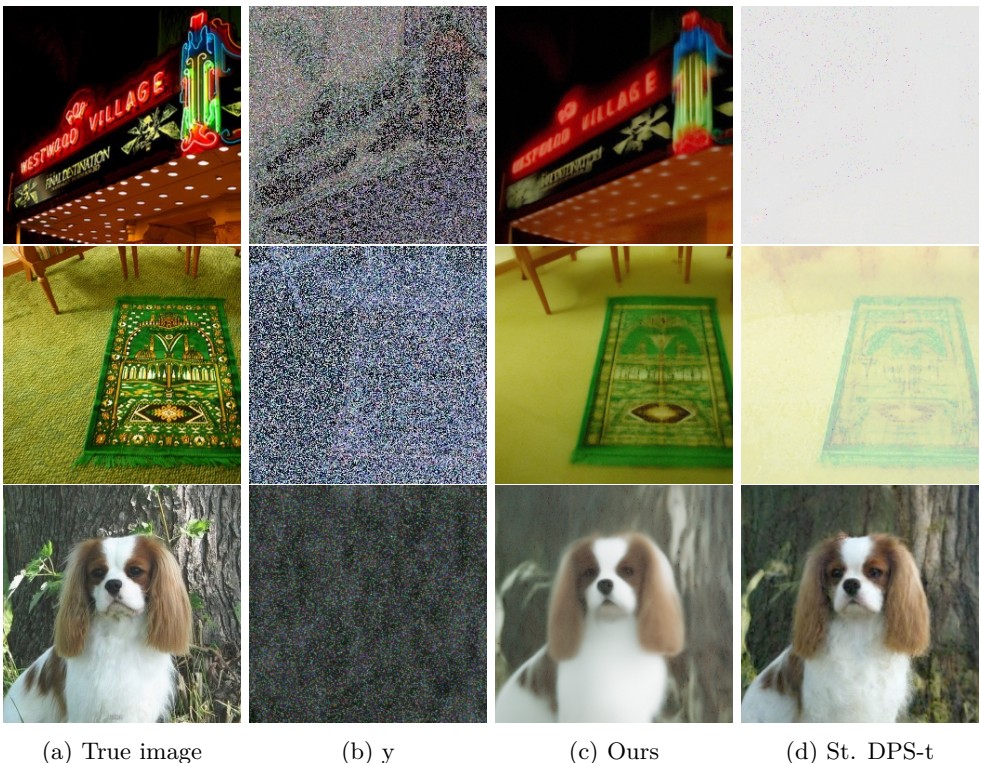

(a) True image      (b) y      (c) Ours      (d) St. DPS-t

Figure 17: Geometric inpainting example ($\alpha = 0.025$) on ImageNet dataset.

