# OpenReview forum: "Score-Based Denoising Diffusion Models for Photon-Starved Image Restoration Problems"
_TMLR — Accepted by TMLR_

### Review · Reviewer_6eck · 2024-10-19

**Summary Of Contributions:**

The authors propose a diffusion based image restoration method using diffusion models to restore low-exposure or photon-starved imaging systems. Their method combines score based diffusion priors with a probabilistic measurement model of images as jointly independent Poisson, Binomial, and Geometric random variables. They modified the existing DiffPIR method for plug-and-play image restoration with a log-likelihood term corresponding to the photon-starved imaging system. Their results show effective measurement conditioned image generation with some improvements in some cases over the state-of-the art methods such as diffusion posterior sampling.

**Audience:**

Yes

**Broader Impact Concerns:**

The improvement over the existing state-of-the-art such as DPS-t is modest, which limits the impact.

**Claims And Evidence:**

Yes

**Requested Changes:**

Section 1 Introduction

" photon-resolving single-photon detectors capable of counting individual photons "

This language is too redundant.


"consists of discrete photon counts, modelled as a realisation of the stochastic
process y|x⋆ ∼ P(α · H(x))"


To be precise, I recommend referring to this as a Poisson distribution, not a Poisson process. A stochastic process refers to a collection of random variables indexed by time or space. See references below

Kao, Edward PC. An introduction to stochastic processes. Courier Dover Publications, 2019.

Todorovic, Petar. An introduction to stochastic processes and their applications. Springer Science & Business Media, 2012.

Gallager, Robert G., and Robert G. Gallager. "Poisson processes." Discrete stochastic processes (1996): 31-55.



In equation (2), it looks like you have assumed the elements of the vector, y, are jointly independent. I recommend making this statement explicitly.


In equation (3), I think understand what you are trying to say, but the language needs to be clarified.

" , the measurement image, y = [y1 . . . , ym] ∈ Nm0 is better modelled as a set of sums of binary detections {0, 1} "

This part is clear, however I would recommend explicitly using the term Bernoulli random variable.

"where Bin(·, ·) stands for the product of independent binomial distribution"

This part is not clear because you are using notation with an argument for the time,  t. A Binomial distribution is defined by the sum of independent Bernoulli trials, but it is not a function of time, t.  If I understand what you are saying correctly, you have a Bernoulli process, taking value zero or one at each time point.  Then, the time integral is a Binomial random variable. However, this Binomial random variable will not be a function of time after integrating over time.

"Note that Eq. 2 is a good approximation of Eq. 3 in practice if α in Eq. 3 is sufficiently small."

Eq (2) is a Poisson log-likelihood and Eq (3) is a distribution. I recommend comparing Eq (1) and Eq (3) or comparing Eq (2) and Eq (4).

Section 2.2


"Denoising diffusion models use Eq. 7 together with weighted score-matching techniques"

Recommend re-phrasing to make this claim about score-based generative models or score-based diffusion models. "Denoising diffusion models" will make the reader assume you mean "Denoising diffusion probabilistic models (DDPMs)" which use a discrete-time framework and do not use score-matching techniques.


Section 3

"That is, we could replace line 4 in Eq. 1..."

I think you mean Algorithm 1 and not Eq. 1


"...with the proximal step associated with f_y, that is..."

Do you need to invoke the notion of a proximal step at all? I think it would be more clear to the reader if you simply justify your method using Bayes rule, breaking a log-posterior into the sum of a log-prior (diffusion based) and log-likelihood (Poisson, Binomial, or Geometric). The notation of "prox" with subsrcipt f_y which is a probability density function, and superscript \rho_t which is a scalar weight on the log-prior is confusing. My recommendation is to remove proximal operator notation from the entire article.



Section 4.1

"Gaussian blur kernel of size 9 × 9 with bandwidth 3-pixels"

The width of a Gaussian kernel is roughly six times the bandwidth, so this implementation will result in a truncated Gaussian kernel.

**Strengths And Weaknesses:**

STRENGTHS
In comparison to most articles on diffusion based image restoration, one strength of this work is the use of a realistic noise model for photon-starved images. The results on measurement conditional image generation are impressive and look better, in some cases, than existing methods.

WEAKNESSES
Both image and quantitative results are only slightly better than the existing state of the art. There were some errors or lack of clarity in the theoretical methods.  The article would benefit from a better explanation of the significance of photon-starved imaging.  It would greatly improve the work to apply the method to some physical measurements from a real photon-starved imaging system.

---

> ### Author Response · Authors · 2024-11-27
> **Reply to Reviewer 6eck**
>
> We sincerely thank the reviewer for their thoughtful comments and valuable suggestions to improve our paper. We also appreciate their acknowledgment that our method employs realistic noise models for photon-starved images, enabling it to generate impressive results that surpass existing methods and SOTA.
>
> (1) We thank the reviewer for acknowledging that both the quantitative and qualitative results produced by our method are better than the existing state of the art. We believe our method is significantly better than SOTA in the following ways:
> - The other methods including the SOTA DPS and DPS-t lack robustness. While they perform comparably to each other and to our method (we often outperform them marginally), they frequently fail and produce poor reconstructions in challenging scenarios. As seen in Table 1, 2 (p.8), 3 (p.10) and Table 4 (p.12), DPS-t led to exploding gradients in motion deblurring with Poisson noise. Both DPS and DPS-t were numerically explosive for most of the testing images in Binomial denoising and Geometric inpainting tasks. In contrast, our method is significantly more robust, as it relies on the appropriate likelihood function, rather than coarse approximations, and on proximal operators which are known to be numerically stable (i.e., maximally monotone firmly non-expansive operators).
> - Even in cases where DPS and DPS-t function properly without the issues of exploding gradients, they tend to produce noticeable artifacts compared to our method, See Figure 2 (p.8).
> - To ensure a fair comparison with DPS and DPS-t, we proposed stabilized versions of both methods to mitigate exploding gradients in Appendix C (p.19). While these stabilized versions show improved performance and fail less frequently, they still produce artifacts compared to our methods. See Figure 9 (e) (p.10) and Figure 12 (g) (p.24), which again highlights the robustness and superiority of our method.
>
> (2) We appreciate the reviewer’s detailed comments on the mathematical formulations and the expressions related to the theoretical aspects of our method. We agree that the concept of the proximal operator is not trivial or familiar to many readers of TMLR. However, it is a cornerstone in modern convex optimization and imaging science literature. We believe that making this concept explicit facilitates the comprehension of our method to these communities, which we are keen to engage.
>
> We will incorporate all the other suggested revisions in the camera-ready manuscript.
>
> (3) We thank the reviewer for their suggestion. In the revised manuscript, we will emphasize more clearly the importance of this new class of imaging technologies that enable remote sensing and computer vision in extreme conditions (e.g. extremely fast regimes, low illumination, underwater, etc.), and mention the many ways in which this disruptive new technology is having a positive impact on society and the economy.
>
> Moreover, we agree that applying our methods to real-world problems would be highly interesting, and this is the main driving force underpinning our research.  However, implementing our approach on real-world data requires substantial modeling effort to develop a forward model suitable for our SPD and SPAD based cameras. This modelling work is currently under investigation and will be reported in future papers.

---

### Review · Reviewer_Abd5 · 2024-11-08

**Summary Of Contributions:**

This paper presents ProxDiffPIR a new Plug-and-Play denoising diffusion model designed for image restoration in extremely low-light, "photon-starved" conditions, where noise statistics are complex due to quantum effects. Unlike previous models, which struggle with high uncertainty and poor regularity in these settings, this approach combines a pre-trained diffusion model with Bayesian inference for adaptable and efficient restoration. Tested on challenging datasets, it outperforms other methods, achieving accurate results.

The proposed method is a direct generalization of DiffPIR, a previous work which is essentially ProxDiffPIR under the assumption that the noise distribution is Gaussian.

**Audience:**

Yes

**Broader Impact Concerns:**

None.

**Claims And Evidence:**

Yes

**Requested Changes:**

No requested changes. The analysis seems complete.

**Strengths And Weaknesses:**

Strengths:

- The paper shows a technique to perform image restoration leveraging diffusion models, that works with generic noise models.
- The experimental results are solid in that the proposed method beats existing ones when measuring PSNR and SSIM. Qualitatively, the image restoration performance of the method is also strong.

Weaknesses:

- From a methodological perspective, the contribution of the work is incremental, as the algorithm is heavily based on DiffPIR. In that sense, the work should be categorized as a computer vision paper rather than a machine learning paper. Although the work falls within the scope of TMLR, it would arguably fit better in a computer vision journal or conference, and it would receive more accurate reviews in such venues.

- What is the reason that the proposed method does not beat DPS with respect to LPIPS?

---

> ### Author Response · Authors · 2024-11-22
> **Reply to Reviewer Abd5**
>
> We would like to thank the reviewer for their thoughtful and constructive feedback, as well as for highlighting that our proposed methodology can be applied to noises stemmed from non-conventional imaging systems, convincingly evidencing that it outperforms the SOTA quantitively and delivers accurate qualitative results.
>
> 1) We agree that our methodology is a generalization of an existing method and that the resulting algorithms are broadly very similar, with a few critical changes. However, we believe that our work is significantly novel in two ways:
>
> a) We advance the PnP paradigm for solving computational imaging problems, a key framework for integrating machine learning and physical models, by proposing the first 	PnP method that can leverage a foundational denoising diffusion model to solve non-	conventional and non-Gaussian imaging problems under photon-starved Poisson, binomial and geometric data.
>
> b) We bring together quantum-enhanced imaging systems and generative diffusion models, two rapidly progressing and deeply impactful research strands that have great 	potential for synergy but have until now been studied separately.
>
> 2) Regarding this latter point, we strongly believe that fostering more interaction between the machine learning and the quantum-enhanced imaging communities will be beneficial for both fields, hence our decision to seek to publish this work in TLMR, as opposed to a computer vision venue. We hope that this piece of research can stimulate a dialogue between these communities and provide an opportunity for the machine learning community to engage with the quantum imaging agenda.
>
> 3) Regarding the performance in terms of LPIPS of both methods, we would first like to underline that the relative LPIPS performance between ProxDiffPIR and DPS depends on the problem considered and the choice of dataset. For example, in Table 1 (p. 8), which depicts the results on the ImageNet dataset, we observe five cases where ProxDiffPIR outperforms or is similar to DPS in LPIPS, and one case where DPS slightly outperforms ProxDiffPIR.
>
> Conversely, in Table 2 (p. 8), which depicts the results on the FFHQ dataset, DPS slightly outperforms ProxDiffPIR in terms of LPIPS by a maximum difference of 0.02 while ProxDiffPIR significantly outperforms DPS in terms of PSNR. We would recall that LPIPS is a perceptual metric. In our experience, it has the tendency to favor images with fine details, even in the presence of excess sharpening artifacts or even noise, so we are not overly concerned about minor improvements or losses in LPIPS performance.
>
> Moreover, by looking at the qualitative results in Figure 3, we observe that DPS tends to produce sharpening artefacts (see for example the teeth zoom-in region), whereas ProxDiffPIR tends to deliver slightly smoother results. As a result, ProxDiffPIR is often significantly better in terms of PSNR performance, at the cost of some deterioration in LPIPS.

---

### Review · Reviewer_g6Hg · 2024-11-17

**Summary Of Contributions:**

Summary:
This paper introduces a plug-and-play denoising diffusion method which aims to tackle photon-starved imaging problems. Because previous studies rely on hand-crafted priors to decide the type of photon noise, this paper aims to tackle the problem in an automated manner. To achieve this goal, the authors extends the foundational DiffPIR to handle non-Gaussian scenarios such as Poisson, Binomial, and Geometric distributions. Further, a proximal optimization step for non-Gaussian log-concave likelihoods is proposed to address non-negativity and non-Lipschitz gradient constrains. Moreover, an efficient DDIM-based diffusion is proposed to reduce computational costs meanwhile maintaining learning performance. Through high-quality visual results and quantitative comparisons, the effectiveness of the proposed method is carefully justified.

Strengths:
- The proposed method is quite adaptive and tackle various situations of photo imaging problems, which is more practical than many other existing works. Plus, the plug-and-play learning manner makes the proposed method easy to apply.
- Rigorous mathematical formulation is provided to demonstrate the proposed method, which strengthens the theoretical credibility of the approach.
- The efficiency improvement is quite impressive, which significantly surpasses other baseline methods.

**Audience:**

Yes

**Broader Impact Concerns:**

No enthic concerns.

**Claims And Evidence:**

Yes

**Requested Changes:**

Please see weaknesses part.

**Strengths And Weaknesses:**

Weaknesses:
- Unclear motivation on improving the effectiveness of photon imaging. Although, this work can tackle various scenarios of problems, but what is the research problem that hinders the performance of existing works is unclear. A further justification on why the proposed method can outperform other baselines should be more clearly presented.
- Lack of computational efficiency comparison with other baseline methods. Although this work conduct test time adaptation which removes the burden of pre-training, the efficiency for each step could still be studied to justify the efficiency.
- Missing several closely-related references:
Huang et al., Robust generalization against photon-limited corruptions via worst-case sharpness minimization, in CVPR 2023
Kang et al., Ddcolor: Towards photo-realistic image colorization via dual decoders, in ICCV 2023
Luo et al., Photo-Realistic Image Restoration in the Wild with Controlled Vision-Language Models, in CVPR 2024.

---

> ### Author Response · Authors · 2024-11-27
> **Reply to Reviewer g6Hg**
>
> (1) We would like to thank the reviewer for their thoughtful remarks, as well as for highlighting that our proposed methodology can tackle forms of noise stemming from quantum imaging systems, that it can be applied straightforwardly by leveraging the plug-and-play paradigm, and that it convincingly delivers outstanding empirical performance advancing SOTA for this class of problems. We are also reassured that the reviewer found the mathematical formulation clear and well motivated.
>
> We also thank the reviewer for pointing out that the original draft was not sufficiently clear regarding the research problem and its intrinsic challenges. Please find below our clarifications which can be included in the camera-ready draft.
>
> As we have mentioned in the introduction, quantum-enhanced imaging is a new class of imaging technologies allowing remote sensing and computer vision systems to operate in extreme conditions (e.g. extremely fast regimes, low illumination, underwater, etc.), underpinned by the development of single-photon sensors such as single-photon detectors (SPDs) and single-photon avalanche diodes (SPADs). These technologies produce measurements that are photon-starved and exhibit challenging noise statistics, such as binomial, geometric, and low-intensity Poisson noise.
>
> Low-photon data present obstacles to the accurate recovery of images for both statistical and computational reasons. From a statistical or inferential perspective, low-photon noise is signal-dependent (i.e. the level of noise is controlled by the intensity of the signal/pixel - this is not the case under additive Gaussian noise processes) and have very poor signal-to-noise ratio, hence the data have very limited information about the solution. This leads to imaging inverse problems with high levels of intrinsic uncertainty. From a computational perspective, the data fidelity terms associated with Poisson, binomial and geometric statistics have poor regularity properties (e.g., non-Lipschitz continuous gradients) and involve positivity constraints on the solution space. This makes it difficult to use gradient-based computation algorithms.
>
> These difficulties hinder the performance of existing strategies. Classical variational methods that are using simplistic and explicit regularization (e.g. total variation flavored norms, l1 and l2 norms) can regularize the estimation problem and capture the structural properties of the image, but this information is not enough to restore fine detail. Data-driven methods, such as PIP and PhD-Net (included as baseline methods), instead of using simplistic regularization, they make use of pre-trained Gaussian denoisers (e.g. BM3D and ResUNet respectively) and improve significantly over the classical variational methods. However, these denoisers are not as effective in capturing fine detail as state-of-the-art generative models, such as the diffusion models. Nonetheless, methods based on diffusion models such as DPS and B-PnP (both included as baseline methods) rely heavily on evaluations of the gradient of the likelihood function. In the class of problems considered, likelihoods have non-regular gradients and discontinuities related to the non-negativity constraints. As a result, both of these methods struggle to deal with low-photon imaging problems and in instabilities occur during the evaluation of the algorithms as a result. On the other hand, ProxDiffPIR is based on proximal optimization, avoids gradient evaluation and ensures that the positivity constraints are satisfied, leading to stable inference for each image. For educational purposes and to make fair comparisons with DPS, we took the initiative to make important algorithmic changes to the original DPS to make it more stable, see Appendices C and D, but still cases of bad convergence can be observed.
>
> (2) Since ProxDiffPIR and DPS were the best performing algorithms in terms of recovering fine details, we included in the original draft computational time comparisons for these methods. Details are included in the table below:
> | ImageNet dataset |ProxDiffPIR|    DPS      |
> |:----------------:|:---------------------:|:------------------:|
> |      Poisson     |  **37** (s/per image) | 520 (s/per image)  |
> |     Binomial     |  **93** (s/per image) |  520 (s/per image) |
> |     Geometric    | **260** (s/per image) |  520 (s/per image) |
>
> |FFHQ dataset|ProxDiffPIR|DPS|
> |:------------:|:--------------------:|:---------------------:|
> |Poisson| **27** (s/per image) |   110 (s/per image)   |
> |Binomial| **80** (s/per image) |   110 (s/per image)   |
> |Geometric|210 (s/per image)  | **110** (s/per image) |
>
> The computational efficiency of ProxDiffPIR depends on the proximal step related to the likelihood and the BFGS step. The above table as well as more details regarding computational time of all the methods can be included in the revised version of the paper.
>
> (3) Thank you, we will be able to add the missing citations in the camera-ready draft.

---

### Decision · Action_Editor_k1KL · 2025-01-02

**Recommendation:** Accept as is

**Comment:**

All reviewers agree that the contribution presented in this paper is valuable, that the approach is clearly motivated and derived, and that the results are compelling. I concur and recommend this paper be accepted, provided the authors address the reviewers' comments in their camera-ready version.

**Audience:**

All reviewers agreed that the contribution presented in this paper is valuable to the subgroups within the TMLR community.

**Claims And Evidence:**

This paper studies image restoration problems in the context of photon-starved conditions, which are becoming increasingly relevant as in quantum-enhanced imaging systems. These are hard inverse problems that involve complex noise statistics, including Poisson, Binomial, and Geometric distributions. In order to employ modern diffusion-based priors, the authors present a generalization of the recent DiffPIR method employing the Plug-and-Play framework by introducing likelihood terms that are not Gaussian and formulating the problem by computing a proximal operation, which is solved numerically. The technique is clearly motivated and presented, and the empirical results demonstrate the advantage of their method against state-of-the-art alternatives.